# Trading Information between Latents in Hierarchical Variational Autoencoders

**Tim Z. Xiao**
University of Tübingen & IMPRS-IS
zhenzhong.xiao@uni-tuebingen.de

**Robert Bamler**
University of Tübingen
robert.bamler@uni-tuebingen.de

## Abstract

Variational Autoencoders (VAEs) were originally motivated (Kingma & Welling, 2014) as probabilistic generative models in which one performs approximate Bayesian inference. The proposal of $\beta$-VAEs (Higgins et al., 2017) breaks this interpretation and generalizes VAEs to application domains beyond generative modeling (e.g., representation learning, clustering, or lossy data compression) by introducing an objective function that allows practitioners to trade off between the information content ("bit rate") of the latent representation and the distortion of reconstructed data (Alemi et al., 2018). In this paper, we reconsider this rate/distortion trade-off in the context of hierarchical VAEs, i.e., VAEs with more than one layer of latent variables. We identify a general class of inference models for which one can split the rate into contributions from each layer, which can then be tuned independently. We derive theoretical bounds on the performance of downstream tasks as functions of the individual layers' rates and verify our theoretical findings in large-scale experiments. Our results provide guidance for practitioners on which region in rate-space to target for a given application.

## 1 Introduction

Variational autoencoders (VAEs) (Kingma & Welling, 2014; Rezende et al., 2014) are a class of deep generative models that are used, e.g., for density modeling (Takahashi et al., 2018), clustering (Jiang et al., 2017), nonlinear dimensionality reduction of scientific measurements (Laloy et al., 2017), data compression (Ballé et al., 2017), anomaly detection (Xu et al., 2018), and image generation (Razavi et al., 2019). VAEs (more precisely, $\beta$-VAEs (Higgins et al., 2017)) span such a diverse set of application domains in part because they can be tuned to a specific task without changing the network architecture, in a way that is well understood from information theory (Alemi et al., 2018).

The original proposal of VAEs (Kingma & Welling, 2014) motivates them from the perspective of generative probabilistic modeling and approximate Bayesian inference. However, the generalization to $\beta$-VAEs breaks this interpretation as they are no longer trained by maximizing a lower bound on the marginal data likelihood. These models are better described as neural networks that are trained to learn the identity function, i.e., to make their output resemble the input as closely as possible. This task is made nontrivial by introducing a so-called (variational) information bottleneck (Alemi et al., 2017; Tishby & Zaslavsky, 2015) at one or more layers, which restricts the information content that passes through these layers. The network activations at the information bottleneck are called latent representations (or simply "latents"), and they split the network into an encoder part (from input to latents) and a decoder part (from latents to output). This separation of the model into an encoder and a decoder allows us to categorize the wide variety of applications of VAEs into three domains:

1. **data reconstruction tasks**, i.e., applications that involve *both the encoder and the decoder*; these include various nonlinear inter- and extrapolations (e.g., image upscaling, denoising, or inpainting), and VAE-based methods for lossy data compression;

2. **representation learning tasks**, i.e., applications that involve *only the encoder*; they serve a downstream task that operates on the (typically lower dimensional) latent representation, e.g., classification, regression, visualization, clustering, or anomaly detection; and

3. **generative modeling tasks**, i.e., applications that involve *only the decoder* are less common but include generating new samples that resemble training data.

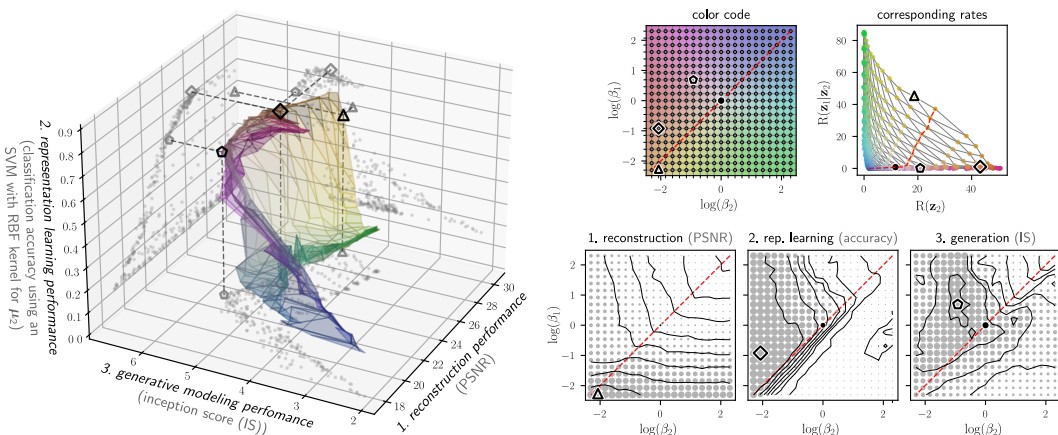

Figure 1: Left: trade-off between performance in the three applications domains of VAEs, using GHVAE trained on the SVHN data set (details: Section 5); higher is better for all three metrics; gray dots on walls show 2d-projections. Right: color code, corresponding layer-wise rates (Eq. 7), and individual performance landscapes (size of dots $\propto$ performance). The hyperparameters $\beta_2$ and $\beta_1$ allow us to tune the HVAE for best data reconstruction ($\triangle$), best representation learning ($\diamond$), or best generative modeling ($\ocircle$). Note that performance landscapes differ strongly across the three applications, and neither a standard VAE ($\beta_2 = \beta_1 = 1$; marked "$\bullet$" in right panels) nor a conventional $\beta$-VAE ($\beta_2 = \beta_1$; dashed red lines) result in optimal models for any of the three applications.

The information bottleneck incentivizes the VAE to encode information into the latents efficiently by removing any redundancies from the input. How agressively this is done can be controlled by tuning the strength $\beta$ of the information bottleneck (Alemi et al., 2018). Unfortunately, information theory distinguishes relevant from redundant information only in a quantitative way that is agnostic to the qualitative features that each piece of information represents about some data point. In practice, many VAE-architectures (Deng et al., 2017; Yingzhen & Mandt, 2018; Ballé et al., 2018) try to separate qualitatively different features into different parts of the latent representation by making the model architecture reflect some prior assumptions about the semantic structure of the data. This allows downstream applications from the three domains discussed above to more precisely target specific qualitative aspects of the data by using or manipulating only the corresponding part of the latent representation. However, in this approach, the degree of detail to which each qualitative aspect is encoded in the latents can be controlled at most indirectly by tuning network layer sizes.

In this paper, we argue both theoretically and empirically that the three different application domains of VAEs identified above require different trade-offs in the amount of information that is encoded in each part of the latent representation. We propose a method to independently control the information content (or "rate") of each layer of latent representations, generalizing the rate/distortion theory of $\beta$-VAEs (Alemi et al., 2018) for VAEs with more than one layer of latents ("hierarchical VAEs" or HVAEs for short). We identify the most general model architecture that is compatible with our proposal and analyze how both theoretical performance bounds and empirically measured performances in each of the above three application domains depend on how rate is distributed across layers.

Our approach is summarized in Figure 1. The 3d-plot shows empirically measured performance metrics (discussed in detail in Section 5.2) for the three application domains identified above. Each point on the colored surface corresponds to different layer-wise rates in an HVAE with two layers of latents. Crucially, the rates that lead to optimal performance are different for each of the three application domains (see markers $\triangle$, $\ocircle$, and $\diamond$ in Figure 1), and none of these three optimal models coincide with a conventional $\beta$-VAE (dashed red lines in right panels). Thus, being able to control each layer's individual rate allows practitioners to train VAEs that target a specific application.

The paper is structured as follows. Section 2 summarizes related work. Section 3 introduces the proposed information-trading method. We then analyze how controlling individual layers' rates can be used to tune HVAEs for specific tasks, i.e., how performance in each of the three application domains identified above depends on the allocation of rates across layers. This analysis is done theoretically in Section 4 and empirically in Section 5. Section 6 provides concluding remarks.

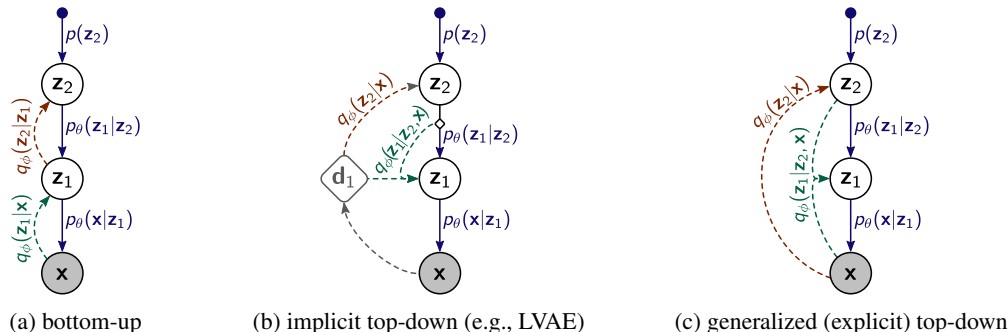

(a) bottom-up       (b) implicit top-down (e.g., LVAE)       (c) generalized (explicit) top-down

Figure 2: Inference (dashed arrows) and generative (solid arrows) models for hierarchical VAEs (HVAEs) with two layers of latent variables. White/gray circles denote latent/observed random variables, respectively; the diamond $\mathbf{d}_1$ in (b) is the result of a deterministic transformation of $\mathbf{x}$.

## 2   RELATED WORK

We group related work into work on model architectures for hierarchical VAEs, and on $\beta$-VAEs.

**Model Design for Hierarchical VAEs.** The original VAE design (Kingma & Welling, 2014; Rezende et al., 2014) has a single layer of latent variables, but recent works (Vahdat & Kautz, 2020; Child, 2021), found that increasing the number of stochastic layers in hierarchical VAEs (HVAEs) improves performance. HVAEs have various designs for their inference models. Sønderby et al. (2016) introduced Ladder VAE (LVAE) with a top-down inference path rather than the naive bottom-up inference (see Section 3), whereas the Bidirectional-Inference VAE (BIVA) (Maaløe et al., 2019) uses a combination of top-down and bottom-up. Our proposed framework applies to a large class of inference models (see Section 3) that includes the popular LVAE (Sønderby et al., 2016).

**$\beta$-VAEs And Their Information-Theoretical Interpretations.** Higgins et al. (2017) introduced an extra hyperparameter $\beta$ in the objective of VAEs that tunes the strength of the information bottleneck, and they observed that large $\beta$ leads to a disentangled latent representation. An information-theoretical interpretation of $\beta$-VAEs was provided in (Alemi et al., 2018) by applying the concept of a (variational) bottleneck (Tishby & Zaslavsky, 2015; Alemi et al., 2017) to autoencoders. Due to this information-theoretical intereration, $\beta$-VAEs are popular models for data compression (Ballé et al., 2017; Minnen et al., 2018; Yang et al., 2020), where tuning $\beta$ allows trading off between the bit rate of compressed data and data distortion. In the present work, we generalize $\beta$-VAEs when applied to HVAEs, and we introduce a framework for tuning the rate of each latent layer individually.

## 3   A HIERARCHICAL INFORMATION TRADING FRAMEWORK

We propose a refinement of the rate/distortion theory of $\beta$-VAEs (Alemi et al., 2018) that admits controlling individual layers' rates in VAEs with more than one layers of latents (hierarchical VAEs).

### 3.1   CONVENTIONAL $\beta$-VAE WITH HIERARCHICAL LATENT REPRESENTATIONS

We consider a hierarchical VAE (HVAE) for data $\boldsymbol{x}$ with $L$ layers of latent representations $\{\boldsymbol{z}_\ell\}_{\ell=1}^L$. Figure 2, discussed further in Section 3.2 below, illustrates various model architectures for the example of $L = 2$. Solid arrows depict the generative model $p_\theta(\{\boldsymbol{z}_\ell\}, \boldsymbol{x})$, where $\theta$ are model parameters (neural network weights). We assume that the implementation factorizes $p_\theta(\{\boldsymbol{z}_\ell\}, \boldsymbol{x})$ as follows,

$$p_\theta(\{\boldsymbol{z}_\ell\}, \boldsymbol{x}) = p_\theta(\boldsymbol{z}_L)\, p_\theta(\boldsymbol{z}_{L-1}|\boldsymbol{z}_L)\, p_\theta(\boldsymbol{z}_{L-2}|\boldsymbol{z}_{L-1}, \boldsymbol{z}_L) \cdots p_\theta(\boldsymbol{z}_1|\boldsymbol{z}_{\geq 2})\, p_\theta(\boldsymbol{x}|\boldsymbol{z}_{\geq 1}) \quad (1)$$

where the notation $\boldsymbol{z}_{\geq n}$ for any $n$ is short for the collection of latents $\{\boldsymbol{z}_\ell\}_{\ell=n}^L$ (thus, $\boldsymbol{z}_{\geq 1}$ and $\{\boldsymbol{z}_\ell\}$ are synonymous), and the numbering of latents from $L$ down to 1 follows the common convention in the literature (Sønderby et al., 2016; Gulrajani et al., 2017; Child, 2021). The loss function of a

normal $\beta$-VAE (Higgins et al., 2017) with this generic architecture would be

$$\mathcal{L}_\beta(\theta, \phi) = \mathbb{E}_{\boldsymbol{x} \sim \mathbb{X}_{\text{train}}} \big[ \underbrace{\mathbb{E}_{q_\phi(\{\boldsymbol{z}_\ell\} | \boldsymbol{x})} \big[ -\log p_\theta(\boldsymbol{x} | \{\boldsymbol{z}_\ell\}) \big]}_{= \text{``distortion''} \, D} + \beta \underbrace{D_{\text{KL}} \big[ q_\phi(\{\boldsymbol{z}_\ell\} | \boldsymbol{x}) \, \| \, p_\theta(\{\boldsymbol{z}_\ell\}) \big]}_{= \text{``rate''} \, R} \big]. \quad (2)$$

Here, $q_\phi(\{\boldsymbol{z}_\ell\} | \boldsymbol{x})$ is the inference (or "encoder") model with parameteres $\phi$, $\mathbb{X}_{\text{train}}$ is the training set, $D_{\text{KL}}[\cdot \| \cdot]$ denotes Kullback-Leibler divergence, and the Lagrange parameter $\beta > 0$ trades off between a (total) rate $R$ and a distortion $D$ (Alemi et al., 2018). Setting $\beta = 1$ turns Eq. 2 into the negative ELBO objective of a regular VAE (Kingma & Welling, 2014). The rate $R$ obtains its name as it measures the (total) information content that $q_\phi$ encodes into the latent representations $\{\boldsymbol{z}_\ell\}$, which would manifest itself in the expected bit rate when one optimally encodes a random draw $\{\boldsymbol{z}_\ell\} \sim q_\phi(\{\boldsymbol{z}_\ell\} | \boldsymbol{x})$ using $p_\theta(\{\boldsymbol{z}_\ell\})$ as an entropy model (Agustsson & Theis, 2020; Bennett et al., 2002). An important observation pointed out in (Alemi et al., 2017) is that, regardless how rate $R$ is traded off against distortion $D$ by tuning $\beta$, their sum $R + D$ is—in expectation under any data distribution $p_{\text{data}}(\boldsymbol{x})$—always lower bounded by the entropy $H[p_{\text{data}}(\boldsymbol{x})] := \mathbb{E}_{p_{\text{data}}(\boldsymbol{x})}[-\log p_{\text{data}}(\boldsymbol{x})]$,

$$\mathbb{E}_{p_{\text{data}}(\boldsymbol{x})}[R + D] \geq H[p_{\text{data}}(\boldsymbol{x})] \qquad \forall \, p_{\text{data}}. \quad (3)$$

**Limitations.** The rate $R$ in Eq. 2 is a property of the *collection* $\{\boldsymbol{z}_\ell\}$ of all latents, which can limit its interpretability for some inference models. For example, the common convention of enumerating layers $\boldsymbol{z}_\ell$ from $\ell = L$ down to 1 in Eq. 1 is reminiscent of a naive architecture for the inference model that factorizes in reverse order compared to Eq. 1 ("bottom up", see dashed arrows in Figure 2(a)), i.e., $q_\phi(\{\boldsymbol{z}_\ell\} | \boldsymbol{x}) = q_\phi(\boldsymbol{z}_1 | \boldsymbol{x}) \, q_\phi(\boldsymbol{z}_2 | \boldsymbol{z}_1) \cdots q_\phi(\boldsymbol{z}_L | \boldsymbol{z}_{L-1})$. Using a HVAE with such a "bottom-up" inference model to reconstruct some given data point $\boldsymbol{x}$ would map $\boldsymbol{x}$ to $\boldsymbol{z}_1$ using $q_\phi(\boldsymbol{z}_1 | \boldsymbol{x})$ and then map $\boldsymbol{z}_1$ back to the data space using $p_\theta(\boldsymbol{x} | \boldsymbol{z}_1)$, thus ignoring all latents $\boldsymbol{z}_\ell$ with $\ell > 1$. Yet, the rate term in Eq. 2 still depends on all latents, including the ones not needed to reconstruct any data (practical VAE-based compression methods using bits-back coding (Frey & Hinton, 1997) would, however, indeed use $\boldsymbol{z}_\ell$ with $\ell > 1$ as auxiliary variables for computational efficiency).

## 3.2 TRADING INFORMATION BETWEEN LATENTS

Many HVAEs used in the literature allow us to resolve the limitations identified in Section 3.1. For example, the popular LVAE architecture (Sønderby et al., 2016), (Figure 2(b)), uses an inference model (dashed arrows) that traverses the latents $\{\boldsymbol{z}_\ell\}$ in the same order as the generative model (solid arrows). We consider the following generalization of this architecture (see Figure 2(c)),

$$q_\phi(\{\boldsymbol{z}_\ell\} | \boldsymbol{x}) = q_\phi(\boldsymbol{z}_L | \boldsymbol{x}) \, q_\phi(\boldsymbol{z}_{L-1} | \boldsymbol{z}_L, \boldsymbol{x}) \, q_\phi(\boldsymbol{z}_{L-2} | \boldsymbol{z}_{L-1}, \boldsymbol{z}_L, \boldsymbol{x}) \cdots q_\phi(\boldsymbol{z}_1 | \boldsymbol{z}_{\geq 2}, \boldsymbol{x}). \quad (4)$$

Formally, Eq. 4 is just the product rule of probability theory and therefore holds for arbitrary inference models $q_\phi(\{\boldsymbol{z}_\ell\} | \boldsymbol{x})$. More practically, however, we make the assumption that the actual implementation of $q_\phi(\{\boldsymbol{z}_\ell\} | \boldsymbol{x})$ follows the structure in Eq. 4. This means that, using the trained model, the most efficient way to map a given data point $\boldsymbol{x}$ to its reconstruction $\hat{\boldsymbol{x}}$ now involves *all* latents $\boldsymbol{z}_\ell$ (either drawing a sample or taking the mode at each step):

$$\boldsymbol{x} \xrightarrow{q_\phi(\boldsymbol{z}_L | \boldsymbol{x})} \boldsymbol{z}_L \xrightarrow{q_\phi(\boldsymbol{z}_{L-1} | \boldsymbol{z}_L, \boldsymbol{x})} \boldsymbol{z}_{L-1} \longrightarrow \cdots \longrightarrow \boldsymbol{z}_2 \xrightarrow{q_\phi(\boldsymbol{z}_1 | \boldsymbol{z}_{\geq 2}, \boldsymbol{x})} \boldsymbol{z}_1 \xrightarrow{p_\theta(\boldsymbol{x} | \{\boldsymbol{z}_\ell\})} \hat{\boldsymbol{x}}. \quad (5)$$

**Layer-wise Rates.** We can interpret Eq. 5 in that it first maps $\boldsymbol{x}$ to a "crude" representation $\boldsymbol{z}_L$, which gets iteratively refined to $\boldsymbol{z}_1$, and finally to a reconstruction $\hat{\boldsymbol{x}}$. Note that each factor $q_\phi(\boldsymbol{z}_\ell | \boldsymbol{z}_{\geq \ell+1}, \boldsymbol{x})$ of the inference model in Eq. 4 is conditioned not only on the previous layers $\boldsymbol{z}_{\geq \ell+1}$ but also on the original data $\boldsymbol{x}$. This allows the inference model to target each refinement step in Eq. 5 such that the reconstruction $\hat{\boldsymbol{x}}$ becomes close to $\boldsymbol{x}$. More formally, we chose the inference architecture in Eq. 4 such that it factorizes over $\{\boldsymbol{z}_\ell\}$ in the same order as the generative model (Eq. 1). This allows us to split the total rate $R$ into a sum of layer-wise rates as follows,

$$R = \mathbb{E}_{q_\phi(\{\boldsymbol{z}_\ell\} | \boldsymbol{x})} \left[ \log \frac{q_\phi(\boldsymbol{z}_L | \boldsymbol{x})}{p_\theta(\boldsymbol{z}_L)} + \log \frac{q_\phi(\boldsymbol{z}_{L-1} | \boldsymbol{z}_L, \boldsymbol{x})}{p_\theta(\boldsymbol{z}_{L-1} | \boldsymbol{z}_L)} + \ldots + \log \frac{q_\phi(\boldsymbol{z}_1 | \boldsymbol{z}_{\geq 2}, \boldsymbol{x})}{p_\theta(\boldsymbol{z}_1 | \boldsymbol{z}_{\geq 2})} \right] \quad (6)$$

$$= R(\boldsymbol{z}_L) + R(\boldsymbol{z}_{L-1} | \boldsymbol{z}_L) + R(\boldsymbol{z}_{L-2} | \boldsymbol{z}_{L-1}, \boldsymbol{z}_L) + \ldots + R(\boldsymbol{z}_1 | \boldsymbol{z}_{\geq 2}).$$

Here,

$$R(\boldsymbol{z}_L) = D_{\text{KL}} \big[ q_\phi(\boldsymbol{z}_L | \boldsymbol{x}) \, \| \, p_\theta(\boldsymbol{z}_L) \big] \qquad \text{and}$$

$$R(\boldsymbol{z}_\ell | \boldsymbol{z}_{\geq \ell+1}) = \mathbb{E}_{q(\boldsymbol{z}_{\geq \ell+1} | \boldsymbol{x})} \big[ D_{\text{KL}} \big[ q_\phi(\boldsymbol{z}_\ell | \boldsymbol{z}_{\geq \ell+1}, \boldsymbol{x}) \, \| \, p_\theta(\boldsymbol{z}_\ell | \boldsymbol{z}_{\geq \ell+1}) \big] \big] \quad (7)$$

quantify the information content of the highest-order latent representation $\boldsymbol{z}_L$ and the (expected) *increase* in information content in each refinement step $\boldsymbol{z}_{\ell+1} \to \boldsymbol{z}_\ell$ in Eq. 5, respectively.

**Controlling Each Layer's Rate.** Using Eqs. 6-7, we generalize the rate/distortion trade-off from Section 3.1 by introducing $L$ individual Lagrange multipliers $\beta_L, \beta_{L-1}, \ldots, \beta_1$, collectively denoted as boldface $\boldsymbol{\beta}$. This leads to a new loss function that generalizes Eq. 2 as follows,

$$\mathcal{L}_{\boldsymbol{\beta}}(\theta, \phi) = \mathbb{E}_{\boldsymbol{x} \sim \mathbb{X}_{\text{train}}}\big[D + \beta_L R(\boldsymbol{z}_L) + \beta_{L-1} R(\boldsymbol{z}_{L-1}|\boldsymbol{z}_L) + \ldots + \beta_1 R(\boldsymbol{z}_1|\boldsymbol{z}_{\geq 2})\big]. \qquad (8)$$

Setting all $\beta$s to the same value recovers the conventional $\beta$-VAE (Eq. 2), which trades off distortion against *total* information content in $\{\boldsymbol{z}_\ell\}$. Tuning each $\beta$-hyperparameter individually allows trading off information content across latents. (In a very deep HVAE (i.e., large $L$) it may be more practical to group layers into only few bins and to use the same $\beta$-value for all layers within a bin.) We analyze how to tune $\beta$s for various applications theoretically in Section 4 and empirically in Section 5.

## 4 INFORMATION-THEORETICAL PERFORMANCE BOUNDS FOR HVAES

In this section, we analyze theoretically how various performance metrics for HVAEs are restricted by the individual layers' rates $R(\boldsymbol{z}_L)$ and $R(\boldsymbol{z}_\ell|\boldsymbol{z}_{\geq \ell+1})$ identified in Eq. 7 for a HVAE with "top-down" inference model. Our analysis motivates the use of the information-trading loss function in Eq. 8 for training HVAEs, following the argument from the introduction that VAEs are commonly used for a vast variety of tasks. As we show, different tasks require different trade-offs that can be targeted by tuning the Lagrange multipliers $\boldsymbol{\beta}$ in Eq. 8. We group tasks into the application domains of (i) data reconstruction and manipulation, (ii) representation learning, and (iii) data generation.

**Data Reconstruction and Manipulation.** The most obvious class of application domains of VAEs includes tasks that combine encoder and decoder to map some data point $\boldsymbol{x}$ to representations $\{\boldsymbol{z}_\ell\}$ and then back to the data space. The simplest performance metric for such data reconstruction tasks is the expected distortion $E_{p_{\text{data}}(\boldsymbol{x})}[D]$, which we can bound by combining Eq. 3 with Eqs. 6-7,

$$\mathbb{E}_{p_{\text{data}}(\boldsymbol{x})}[D] \geq H[p_{\text{data}}(\boldsymbol{x})] - \mathbb{E}_{p_{\text{data}}(\boldsymbol{x})}\big[R(\boldsymbol{z}_L) + R(\boldsymbol{z}_{L-1}|\boldsymbol{z}_L) + \cdots + R(\boldsymbol{z}_1|\boldsymbol{z}_{\geq 2})\big]. \qquad (9)$$

Eq. 9 would suggest that higher rates (i.e., lower $\beta$'s) are always better for data reconstruction tasks. However, in many practical tasks (e.g., image upscaling, denoising, or inpainting) the goal is not solely to reconstruct the original data but also to manipulate the latent representations $\{\boldsymbol{z}_\ell\}$ in a meaningful way. Here, lower rates can lead to more semantically meaningful representation spaces (see, e.g., Section 5.6 below). Controlling how rate is distributed across layers via Eq. 8 may allow practitioners to have a semantically meaningful high-level representation $\boldsymbol{z}_L$ with low rate $R(\boldsymbol{z}_L)$ while still retaining a high *total* rate $R$, thus allowing for low distortion $D$ without violating Eq. 9.

**Representation Learning.** In many practical applications, VAEs are used as nonlinear dimensionality reduction methods to prepare some complicated high-dimensional data $\boldsymbol{x}$ for downstream tasks such as classification, regression, visualization, clustering, or anomaly detection. We consider a classifier $p_{\text{cls.}}(y|\boldsymbol{z}_\ell)$ operating on the latents $\boldsymbol{z}_\ell$ at some level $\ell$. We assume that the (unknown) true data generative process $p_{\text{data}}(y, \boldsymbol{x}) = p_{\text{data}}(y) \, p_{\text{data}}(\boldsymbol{x}|y)$ generates data $\boldsymbol{x}$ conditioned on some true label $y$, thus defining a Markov chain $y \xrightarrow{p_{\text{data}}} \boldsymbol{x} \xrightarrow{q_\phi} \boldsymbol{z}_\ell \xrightarrow{p_{\text{cls.}}} \hat{y}$ where $\hat{y} := \arg\max_y p_{\text{cls.}}(y|\boldsymbol{z}_\ell)$. Classification accuracy is bounded (Meyen, 2016) by a function of the mutual information $I_q(y; \boldsymbol{z}_\ell)$,

$$I_q(y; \boldsymbol{z}_\ell) \leq I_q(\boldsymbol{x}; \boldsymbol{z}_\ell) \equiv \mathbb{E}_{p_{\text{data}}(\boldsymbol{x})}\left[\mathbb{E}_{q_\phi(\boldsymbol{z}_\ell|\boldsymbol{x})}\left[\log \frac{q_\phi(\boldsymbol{z}_\ell|\boldsymbol{x})}{q_\phi(\boldsymbol{z}_\ell)}\right]\right] \qquad (10)$$

$$= \mathbb{E}_{p_{\text{data}}(\boldsymbol{x})}\left[\mathbb{E}_{q_\phi(\boldsymbol{z}_\ell|\boldsymbol{x})}\left[\log \frac{q_\phi(\boldsymbol{z}_\ell|\boldsymbol{x})}{p_\theta(\boldsymbol{z}_\ell)}\right]\right] - D_{\text{KL}}\big[q_\phi(\boldsymbol{z}_\ell) \,\|\, p_\theta(\boldsymbol{z}_\ell)\big]$$

$$\leq \mathbb{E}_{p_{\text{data}}(\boldsymbol{x})}\left[\mathbb{E}_{q_\phi(\boldsymbol{z}_{\geq \ell}|\boldsymbol{x})}\left[\log \frac{q_\phi(\boldsymbol{z}_{\geq \ell}|\boldsymbol{x})}{p_\theta(\boldsymbol{z}_{\geq \ell})}\right]\right.$$

$$\left. - \mathbb{E}_{q_\phi(\boldsymbol{z}_\ell|\boldsymbol{x})}\Big[D_{\text{KL}}\big[q_\phi(\boldsymbol{z}_{\geq \ell+1} \,|\, \boldsymbol{x}, \boldsymbol{z}_\ell) \,\|\, p_\theta(\boldsymbol{z}_{\geq \ell+1}|\boldsymbol{z}_\ell)\big]\Big]\right]$$

$$\leq \mathbb{E}_{p_{\text{data}}(\boldsymbol{x})}\big[\underbrace{R(\boldsymbol{z}_L) + R(\boldsymbol{z}_{L-1}|\boldsymbol{z}_L) + \ldots + R(\boldsymbol{z}_\ell \,|\, \boldsymbol{z}_{\geq \ell+1})}_{=: R(\boldsymbol{z}_{\geq \ell}) \,(\leq R)}\big].$$

Here, $q_\phi(\boldsymbol{z}_\ell) := \mathbb{E}_{p_{\text{data}}(\boldsymbol{x})}[q_\phi(\boldsymbol{z}_\ell|\boldsymbol{x})]$ and we identify $R(\boldsymbol{z}_{\geq \ell})$ as the rate accumulated in all layers from $\boldsymbol{z}_L$ to $\boldsymbol{z}_\ell$. The first inequality in Eq. 10 comes from the data processing inequality (MacKay,

2003), and the other two inequalities result from discarding the (nonnegative) KL-terms. The classification accuracy is thus bounded by (Meyen, 2016) (see also proof in Appendix B)

$$\text{class. accuracy} \leq f^{-1}\big(I_q(y; \boldsymbol{z}_\ell)\big) \leq f^{-1}\big(\mathbb{E}_{p_{\text{data}}(\boldsymbol{x})}[R(\boldsymbol{z}_{\geq \ell})]\big) \quad \big(\leq f^{-1}\big(\mathbb{E}_{p_{\text{data}}(\boldsymbol{x})}[R]\big)\big) \quad (11)$$

where $f^{-1}$ is the inverse of the monotonic function $f(\alpha) = H[p_{\text{data}}(y)] + \alpha \log \alpha + (1-\alpha) \log \frac{1-\alpha}{M-1}$ with $M$ being the number of classes and $H[p_{\text{data}}(y)] \leq \log M$ the marginal label entropy. Eq. 11 suggests that the accuracy of an optimal classifier on $\boldsymbol{z}_\ell$ would increase as the rate $R(\boldsymbol{z}_{\geq \ell})$ accumulated from $\boldsymbol{z}_L$ to $\boldsymbol{z}_\ell$ grows (i.e., as $\beta_{\geq \ell} \to 0$), and that the rate added in downstream layers $\boldsymbol{z}_{<\ell}$ would be irrelevant. Practical classifiers, however, have a limited expressiveness, which a very high rate $R(\boldsymbol{z}_{\geq \ell})$ might exceed by encoding too many details into $\boldsymbol{z}_\ell$ that are not necessary for classification. We observe in Section 5.6 that, in such cases, increasing the rates of *downstream* layers $\boldsymbol{z}_{<\ell}$ improves classification accuracy as it allows keeping $\boldsymbol{z}_\ell$ simpler by deferring details to $\boldsymbol{z}_{<\ell}$.

**Data Generation.** The original proposal of VAEs (Kingma & Welling, 2014) motivated them from a generative modeling perspective using that, for $\beta = 1$, the negative of the loss function in Eq. 2 is a lower bound on the log marginal data likelihood. This suggests setting all $\beta$-hyperparameters in Eq. 8 to values close to 1 if a HVAE is used primarily for its generative model $p_\theta$.

In summary, our theoretical analysis suggests that optimally tuned layer-wise rates depend on whether a HVAE is used for data reconstruction, representation learning, or data generation. The next section tests our theoretical predictions empirically for the same three application domains.

## 5 EXPERIMENTS

To demonstrate the features of our hierarchical information trading framework, we run large-scale grid searches over a two-dimensional rate space using two different implementations of HVAEs and three different data sets. Although the proposed framework is applicable for HVAEs with $L \geq 2$, we only use HVAEs with $L = 2$ in our experiments for simplicity and visualization purpose.

### 5.1 EXPERIMENTAL SETUP

**Data sets.** We used the SVHN (Netzer et al., 2011) and CIFAR-10 (Krizhevsky, 2009) data sets (both $32 \times 32$ pixel color images), and MNIST (LeCun et al., 1998) ($28 \times 28$ binary pixel images). SVHN consists of photographed house numbers from 0 to 9, which are geometrically simpler than the 10 classes of objects from CIFAR-10 but more complex than MNIST digits. Most results shown in the main paper use SVHN; comprehensive results for CIFAR-10 and MNIST are shown in Appendix A.2 and tell a similar story except where explicitly discussed.

**Model Architectures.** For the generative model (Eq. 1), we assume a (fixed) standard Gaussian prior $p(\boldsymbol{z}_2) = \mathcal{N}(\mathbf{0}, \mathbf{I})$, and we use diagonal Gaussian models for $p_\theta(\boldsymbol{z}_1|\boldsymbol{z}_2) = \mathcal{N}(g_\mu(\boldsymbol{z}_2), g_\sigma(\boldsymbol{z}_2)^2)$ and (for SVHN and CIFAR-10) $p_\theta(\boldsymbol{x}|\boldsymbol{z}_1) = \mathcal{N}(g_{\mu'}(\boldsymbol{z}_1), \sigma_{\boldsymbol{x}}^2 \mathbf{I})$ (this is similar to, e.g., (Minnen et al., 2018)). Here, $g_\mu$, $g_\sigma$, and $g_{\mu'}$, denote neural networks (see details below). Since MNIST has binary pixel values, we model it with a Bernoulli distribution for $p_\theta(\boldsymbol{x}|\boldsymbol{z}_1) = \text{Bern}(g_{\mu'}(\boldsymbol{z}_1))$. For the inference model, we also use diagonal Gaussian models for $q_\phi(\boldsymbol{z}_2|\boldsymbol{x}) = \mathcal{N}(f_\mu(\boldsymbol{x}), f_\sigma(\boldsymbol{x})^2)$ and for $q_\phi(\boldsymbol{z}_1|\boldsymbol{x}, \boldsymbol{z}_2) = \mathcal{N}(f_{\mu'}(\boldsymbol{x}, \boldsymbol{z}_2), f_{\sigma'}(\boldsymbol{x}, \boldsymbol{z}_2)^2)$, where $f_\mu$, $f_\sigma$, $f_{\mu'}$, and $f_{\sigma'}$ are again neural networks.

We examine both LVAE (Figure 2(b)) and our generalized top-down HVAEs (GHVAEs; see Figure 2(c)), using simple network architectures with only 2 to 3 convolutional and 1 fully connected layers (see Appendix A.1 for details) so that we can scan a large rate-space efficiently. Note that we are not trying to find the new state-of-the-art HVAEs. Results for LVAE are in Appendix A.2.2.

We trained 441 different HVAEs for each data set/model combination, scanning the rate-hyperparameters $(\beta_2, \beta_1)$ over a $21 \times 21$ grid ranging from 0.1 to 10 on a log scale in both directions (see Figure 1 on page 2, right panels). Each model took about 2 hours to train on an RTX-2080Ti GPU ($\sim 27$ hours in total for each data set/model combination using 32 GPUs in parallel).

**Baselines.** Our proposed framework (Eq. 8) generalizes over both VAEs and $\beta$-VAEs (Eq. 2), which we obtain in the cases $\beta_2 = \beta_1 = 1$ and $\beta_2 = \beta_1$, respectively. These baselines are indicated as black "o" and red "o" circles, respectively, in Figures 3, 5, 6, and 7, discussed below.

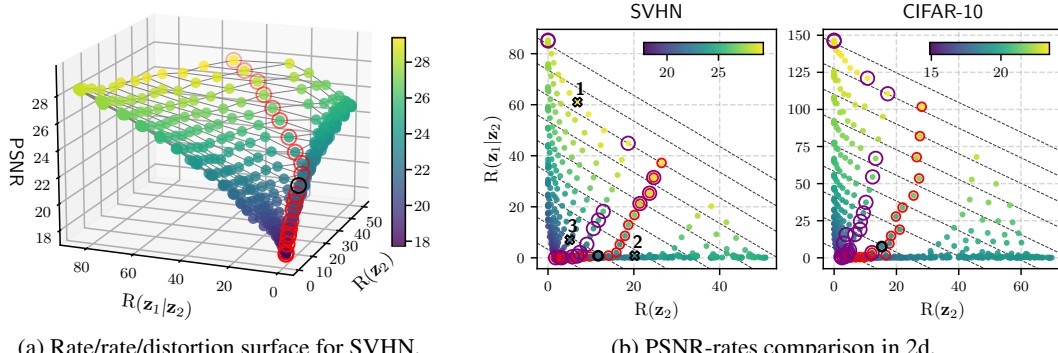

(a) Rate/rate/distortion surface for SVHN.          (b) PSNR-rates comparison in 2d.

Figure 3: PSNR-rate trade-off for GHVAEs trained on SVHN and CIFAR-10. Figure (a) visualizes the same data as the left panel of (b) in 3d. Black circles "o" mark standard VAEs ($\beta_2 = \beta_1 = 1$), red circles "o" mark $\beta$-VAEs ($\beta_2 = \beta_1$), and purple circles "o" mark optimal models along constant total rate (dashed diagonal lines) as defined in Section 5.3. Crosses point to columns in Figure 4.

**Metrics.** Performance metrics for the three application domains of VAEs mentioned in the introduction are introduced at the beginnings of the corresponding Sections 5.4-5.6. In addition, we evaluate the individual rates $R(\mathbf{z}_2)$ and $R(\mathbf{z}_1|\mathbf{z}_2)$ (Eq. 7), which we report in *nats* (i.e., to base $e$).

## 5.2    THERE IS NO "ONE HVAE FITS ALL"

Figure 1 on page 2 summarizes our results. The $21 \times 21$ GHVAEs trained with the grid of hyperparameters $\beta_2$ and $\beta_1$ map out a surface in a 3d-space spanned by suitable metrics for the three application domains (metrics defined in Sections 5.4-5.6 below). The two upper right panels map colors on this surface to $\beta$s used for training and to the resulting layer-wise rates, respectively. The lower right panels show performance landscapes and identify the optimal models for the three application domains of data reconstruction ($\triangle$), representation learning ($\diamond$), and generative modeling ($\bigcirc$).

The figure shows that moving away from a conventional $\beta$-VAE ($\beta_2 = \beta_1$; dashed red lines in Figure 1) allows us to find better models for a given application domain as the three application domains favor vastly different regions in $\beta$-space. Thus, *there is no single HVAE that is optimal for all tasks*, and a HVAE that has been optimized for one task can perform poorly on a different task.

## 5.3    DEFINITION OF THE OPTIMAL MODEL FOR A GIVEN TOTAL RATE

One of the questions we study in Sections 5.4-5.6 below is: "Which allocation of rates across layers results in best model performance *if we keep the total rate $R$ fixed*". Unfortunately, it is difficult to keep $R$ fixed at training time since we control rates only indirectly via their Lagrange multipliers $\beta_2$ and $\beta_1$. We instead use the following definition, illustrated in Figure 6 for a performance metric introduced in Section 5.6 below. The figure plots the performance metric over $R$ for all $21 \times 21$ $\beta$-settings and highlights with purple circles "o" all points on the upper convex hull. These highlighted models are optimal for a small interval of total rates in the following sense: if we use the total rates $R$ of all "o" to partition the horizontal axis into intervals then, by definition of the convex hull, each "o" represents the model with highest performance in either the interval to its left or the one to its right.

## 5.4    PERFORMANCE ON DATA RECONSTRUCTION

Reconstruction is a popular task for VAEs, e.g., in the area of lossy compression (Ballé et al., 2017). We measure reconstruction quality using the common peak signal-to-noise ratio (PSNR), which is equal to $\mathbb{E}_{\boldsymbol{x} \sim \mathbb{X}_{\text{test}}}[-\log D]$ up to rescaling and shifting. Higher PSNR means better reconstruction.

Figure 3(a) shows a 3d-plot of PSNR as a function of both $R(\boldsymbol{z}_1|\boldsymbol{z}_2)$ and $R(\boldsymbol{z}_2)$ for SVHN, thus generalizing the rate/distortion curve of a conventional $\beta$-VAE to a rate/rate/distortion surface. Figure 3(b) introduces a more compact 2d-representation of the same data that we use for all remaining metrics in the rest of this section and in Appendix A.2, and it also shows results for CIFAR-10.

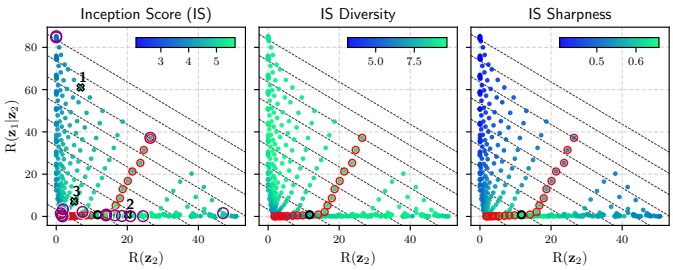
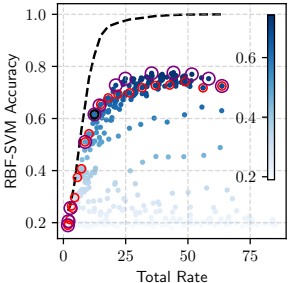

Figure 4: Samples (top) and reconstructions (bottom) from 3 different models (blue column labels "1", "2", and "3" from left to right correspond to crosses "1", "2", and "3" in Figures 3(b) & 5). Consistent with PSNR and IS metrics, model "1" produces poorest samples but best reconstructions.

Figure 5: Sample generation performance, measured in Inception Score (IS, see Eq. 12) and its factorization into diversity and sharpness as a function of layer-wise rates for GHVAEs trained using SVHN data. Crosses in left panel correspond to samples shown in Figure 4. Markers "o", "o", and "o" same as in Figure 3.

Figure 6: RBF-SVM classification accuracies on $\boldsymbol{\mu}_2$. Dashed line shows theoretical bound (Eq. 11). Other markers as in Figure 3.

Unsurprisingly and consistent with Eq. 9, reconstruction performance improves as total rate grows. However, minimizing distortion without any constraints is not useful in practice as we can simply use the original data, which has no distortion. To simulate a practical constraint in, e.g., a data-compression application, we consider models with optimal PSNR *for a given total rate $R$* (as defined in Section 5.3) which are marked as purple circles "o" in Figure 3(b). We see for both SVHN and CIFAR-10 that conventional $\beta$-VAEs ($\beta_2 = \beta_1$; red circles) perform somewhat suboptimal for a given total rate and can be improved by trading some rate in $\boldsymbol{z}_2$ for some rate in $\boldsymbol{z}_1$. Reconstruction examples for the three models marked with crosses in Figure 3(b) are shown in Figure 4 (bottom). Visual reconstruction quality improves from "3" to "2" to "1", consistent with reported PSNRs.

## 5.5 PERFORMANCE ON SAMPLE GENERATION

We next evaluate how tuning layer-wise rates affects the quality of samples from the generative model. We measure sample quality by the widely used Inception Score (IS) (Salimans et al., 2016),

$$\text{IS} = \exp\left\{\mathbb{E}_{p_\theta(\boldsymbol{x})}\left[D_{\text{KL}}\left[p_{\text{cls.}}(y|\boldsymbol{x}) \| p_{\text{cls.}}(y)\right]\right]\right\} = e^{H[p_{\text{cls.}}(y)]} \times e^{-\mathbb{E}_{p_\theta(\boldsymbol{x})}[H[p_{\text{cls.}}(y|\boldsymbol{x})]]} \quad (12)$$

Here, $p_\theta$ is the trained generative model (Eq. 1), $p_{\text{cls.}}(y|\boldsymbol{x})$ is the predictive distribution of a classifier trained on the same training set, and $p_{\text{cls.}}(y) := \mathbb{E}_{p_\theta(\boldsymbol{x})}[p_{\text{cls.}}(y|\boldsymbol{x})]$. The second equality in Eq. 12 follows Barratt & Sharma (2018) to split IS into a product of a diversity score and a sharpness score. Higher is better for all scores. The classifier is a ResNet-18 (He et al., 2016) for SVHN (test accuracy 95.02 %) and a DenseNet-121 (Huang et al., 2017) for CIFAR-10 (test accuracy 94.34 %).

Figure 5 (left) shows IS for GHVAEs trained on SVHN. Unlike the results for PSNR, here, higher rate does not always lead to better sample quality: for very high $R(\boldsymbol{z}_2)$ and low $R(\boldsymbol{z}_1|\boldsymbol{z}_2)$, IS eventually drops. The region of high IS is in the area where $\beta_2 < \beta_1$, i.e., where $R(\boldsymbol{z}_2)$ is higher than in a comparable conventional $\beta$-VAE. The center and right panels of Figure 5 show diversity and sharpness, indicating that IS is mainly driven here by sharpness, which depends mostly on $R(\boldsymbol{z}_2)$, possibly because $\boldsymbol{z}_2$ captures higher-level concepts than $\boldsymbol{z}_1$ that may be more important to the classifier in Eq. 12. Samples from the the three models marked with crosses in Figure 5 are shown in Figure 4 (top). Visual sample quality improves from "1" to "3" to "2", consistent with reported IS.

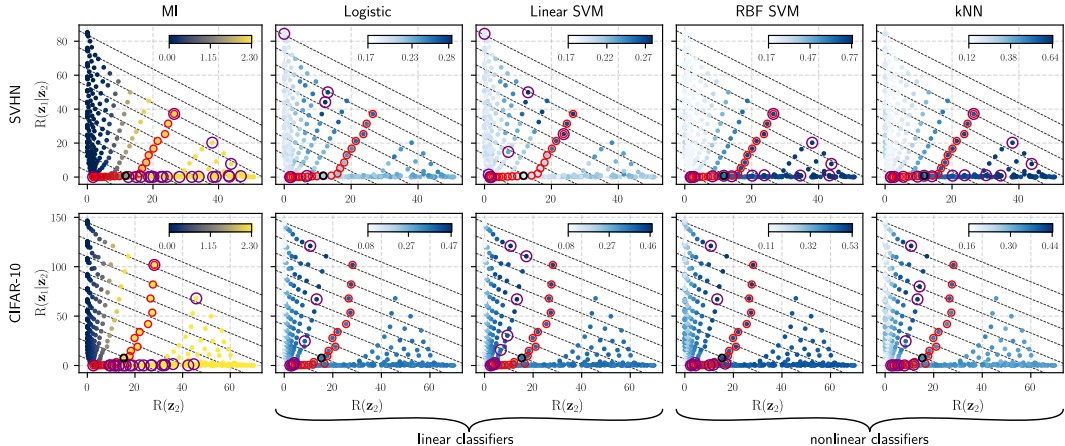

Figure 7: Mutual information (MI) $I_q(y; z_2)$ and classification accuracies of four classifiers (see column labels) as a function of layer-wise rates $R(z_2)$ and $R(z_1|z_2)$. Classifiers are conditioned on $\mu_2 := \arg\max_{z_2} q(z_2|x)$ learned from GHVAEs trained with SVHN (top) and CIFAR-10 (bottom). Markers "○", "○", and "○" same as in Figure 3.

## 5.6 PERFORMANCE ON REPRESENTATION LEARNING FOR DOWNSTREAM CLASSIFICATION

VAEs are very popular for representation learning as they map complicated high dimensional data $x$ to typically lower dimensional representations $\{z_\ell\}$. To measure the quality of learned representations, we train two sets of classifiers on a labeled test set for each trained HVAE, each consisting of: logistic regression, a Support Vector Machine (SVM) (Boser et al., 1992) with linear kernel, an SVM with RBF kernel, and $k$-nearest neighbors (kNN) with $k = 5$. One set of classifiers is conditioned on the mode $\mu_2$ of $q_\phi(z_2|x)$ and the other one on the mode $\mu_1$ of $q_\phi(z_1|z_2, x)$, where $z_2 \sim q_\phi(z_2|x)$. We use the implementations from scikit-learn (Pedregosa et al., 2011) for all classifiers.

Figure 7 shows the classification accuracies (columns 2-5) for all classifiers trained on $\mu_2$. The first column shows the mutual information $I_q(y; z_2)$, which depends mainly on $R(z_2)$ as expected from Eq. 10. As long as the classifier is expressive enough (e.g., RBF-SVM or kNN) and the data set is simple (SVHN; top row), higher mutual information ($\approx$ higher $R(z_2)$) corresponds to higher classification accuracies, consistent with Eq. 11. But for less expres-

Table 1: Optimal classification accuracies (across all $(\beta_2, \beta_1)$-settings) using either $\mu_2$ or $\mu_1$.

| Data Set | log. reg. | lin. SVM | RBF SVM | kNN |
|---|---|---|---|---|
| SVHN ($\mu_2$) | 28.43 % | 27.87 % | **77.60 %** | **64.25 %** |
| SVHN ($\mu_1$) | **45.77 %** | **49.81 %** | 59.28 % | 56.49 % |
| CIFAR-10 ($\mu_2$) | **47.36 %** | **46.95 %** | **53.15 %** | **44.20 %** |
| CIFAR-10 ($\mu_1$) | 43.27 % | 42.55 % | 45.60 % | 39.25 % |

sive (e.g., linear) classifiers or more complex data (CIFAR-10; bottom row), increasing $R(z_1|z_2)$ improves classification accuracy (see purple circles "○" in corresponding panels), consistent with the discussion below Eq. 11. We see a similar effect (Table 1) for most classifier/data set combinations when replacing $\mu_2$ by $\mu_1$, which has more information about $x$ but is also higher dimensional.

## 6 CONCLUSIONS

We classified the various tasks that can be performed with Variational Autoencoders (VAEs) into three application domains and argued that each domain has different trade-offs, such that a good VAE for one domain is not necessarily good for another. This observation motivated us to propose a refinement of the rate/distortion theory of VAEs that allows trading off rates across individual layers of latents in hierarchical VAEs. We showed both theoretically and empirically that the proposal indeed provides practitioners better control for tuning VAEs for the three application domains. In the future, it would be interesting to explore adaptive schedules for the Lagrange parameters $\beta$ that would make it possible to target a specific given rate for each layer in a single training run, for example by using the method proposed by Rezende & Viola (2018).

ACKNOWLEDGMENTS

The authors would like to thank Johannes Zenn, Zicong Fan, Zhen Liu for their helpful discussion. Funded by the Deutsche Forschungsgemeinschaft (DFG, German Research Foundation) under Germany's Excellence Strategy – EXC number 2064/1 – Project number 390727645. This work was supported by the German Federal Ministry of Education and Research (BMBF): Tübingen AI Center, FKZ: 01IS18039A. The authors thank the International Max Planck Research School for Intelligent Systems (IMPRS-IS) for supporting Tim Z. Xiao.

**Reproducibility Statement.** All code necessary to reproduce the results in this paper is available at https://github.com/timxzz/HIT/.

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

## A Experiment Supplementaries

### A.1 Implementation Details

Table 2: Model architecture details for generalized top-down HVAEs (GHVAEs) used in Section 5. *Conv* and *ConvTransp* denote the convolutional and transposed convolutional layer, which has the corresponding input: *input channel, output channel, kernel size, stride, padding*. *FC* represents fully connected layer.

| Data set | $q(z_2|x)$ | $q(z_1|z_2, x)$ | $p(z_1|z2)$ | $p(x|z_1)$ |
|---|---|---|---|---|
| SVHN/ CIFAR-10 | *Share:* Conv(3, 32, 4, 2, 1), Conv(32, 32, 4, 2, 1), Conv(32, 32, 4, 2, 1) 
 *For mean:* FC(In=512, Out=32) 
 *For variance:* FC(In=512, Out=32) | *For x:* Conv(3, 32, 4, 2, 1), Conv(32, 32, 4, 2, 1) 
 *For $z_2$:* FC(In=32, Out=512) 
 *Share:* ConvTransp(64, 32, 4, 2, 1) 
 *For mean:* Conv(32, 32, 3, 1, 1) 
 *For variance:* Conv(32, 32, 3, 1, 1) | *Share:* FC(In=32, Out=256) 
 *For mean:* FC(In=256, Out=512) 
 *For variance:* FC(In=256, Out=512) | ConvTransp(32, 32, 4, 2, 1), ConvTransp(32, 32, 4, 2, 1), ConvTransp(32, 3, 4, 2, 1) |
| | $z_1$ dims: 512 | $z_2$ dims: 32 | $\sigma_x = 0.71$ | Total params: 475811 |
| MNIST (Binary) | *Share:* Conv(1, 16, 4, 2, 1), Conv(16, 16, 4, 2, 1), Conv(16, 16, 4, 1, 0) 
 *For mean:* FC(In=256, Out=20) 
 *For variance:* FC(In=256, Out=20) | *For x:* Conv(1, 16, 4, 2, 1), Conv(16, 16, 4, 1, 0) 
 *For $z_2$:* FC(In=20, Out=256) 
 *Share:* ConvTransp(32, 16, 4, 1, 0) 
 *For mean:* Conv(16, 16, 3, 1, 1) 
 *For variance:* Conv(16, 16, 3, 1, 1) | *Share:* FC(In=20, Out=128) 
 *For mean:* FC(In=128, Out=256) 
 *For variance:* FC(In=128, Out=256) | ConvTransp(16, 16, 4, 1, 0), ConvTransp(16, 16, 4, 2, 1), ConvTransp(16, 1, 4, 2, 1) |
| | $z_1$ dims: 256 | $z_2$ dims: 20 | $\sigma_x$: N/A | Total params: 122713 |

Table 3: Model architecture details for LVAEs used in Section 5. *Conv* and *ConvTransp* denote the convolutional and transposed convolutional layer, which has the corresponding input: *input channel, output channel, kernel size, stride, padding*. *FC* represents fully connected layer.

| Data set | $q(z_2|x)$ | $q(z_1|z_2, x)$ | $p(z_1|z2)$ | $p(x|z_1)$ |
|---|---|---|---|---|
| SVHN/ CIFAR-10 | *Share:* Conv(3, 32, 4, 2, 1), Conv(32, 32, 4, 2, 1), Conv(32, 32, 4, 2, 1) 
 *For mean:* FC(In=512, Out=32) 
 *For variance:* FC(In=512, Out=32) | *Involve* d*:* Conv(32, 32, 4, 2, 1) 
 *For mean:* Conv(32, 32, 3, 1, 1) 
 *For variance:* Conv(32, 32, 3, 1, 1) | *Share:* FC(In=32, Out=256) 
 *For mean:* FC(In=256, Out=512) 
 *For variance:* FC(In=256, Out=512) | ConvTransp(32, 32, 4, 2, 1), ConvTransp(32, 32, 4, 2, 1), ConvTransp(32, 3, 4, 2, 1) |
| | $z_1$ dims: 512 | $z_2$ dims: 32 | $\sigma_x = 0.71$ | Total params: 408131 |

## A.2 ADDITIONAL RESULTS

Here we attached the results for MNIST, as well as the full results for LVAE on SVHN and generalized top-down HVAEs on CIFAR-10.

### A.2.1 RESULTS FOR GENERALIZED TOP-DOWN HVAES ON MNIST

We also evaluate our proposed framework using generalized top-down HVAEs trained on binary MNIST data (i.e., black and white images rather than grayscale).

We note that the inception score (IS) behaves different in our MNIST models compared to SVHN (see Figure 5) in that optimal IS in MNIST occurs for high $R(z_1|z_2)$ rather than high $R(z_2)$. This indicates that semantically low-level properties (hand-writing style) of MNIST might have more variation than high level properties (the digit), whereas SVHN images show variation in additional high-level properties such as the background color.

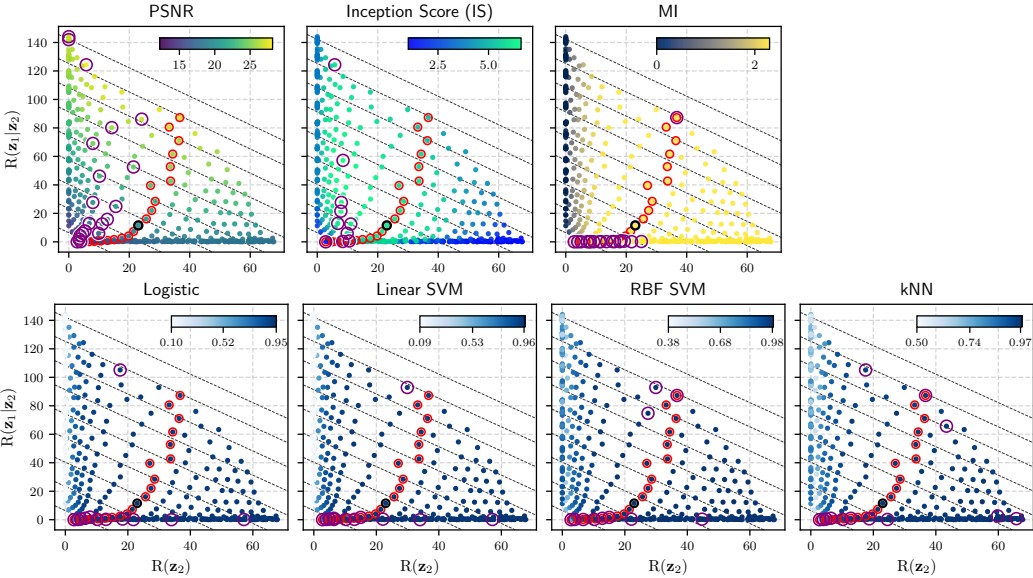

Figure 8: Trade-offs between rates and all metrics we used in Section 5 from the generalized top-down HVAEs trained with MNIST. The results from the standard VAE (i.e. $\beta_2 = \beta_1 = 1$) and the $\beta$-VAE (i.e. $\beta_2 = \beta_1$) are marked with "○" and "○". The markers "○" highlight the optimal models selected using convex hull (see Figure 6 for details). The diagonal grid lines are references for equivalent total rates, i.e. points on the same line have the same total rates.

### A.2.2 RESULTS FOR LVAE ON SVHN

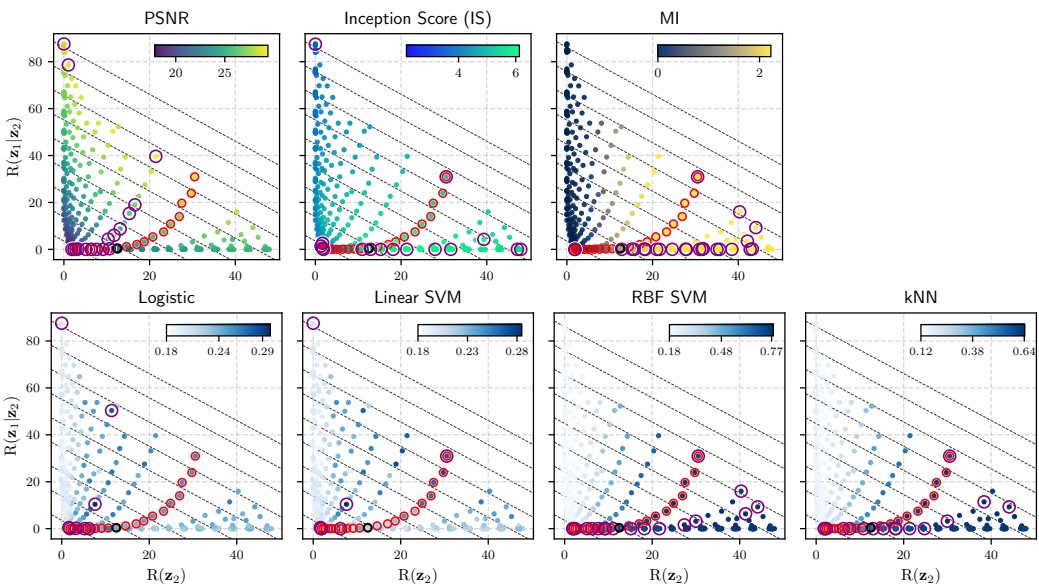

Figure 9: Trade-offs between rates and all metrics we used in Section 5 from LVAE trained with SVHN. The results from the standard VAE (i.e. $\beta_2 = \beta_1 = 1$) and the $\beta$-VAE (i.e. $\beta_2 = \beta_1$) are marked with "o" and "o". The markers "o" highlight the optimal models selected using convex hull (see Figure 6 for details). The diagonal grid lines are references for equivalent total rates, i.e. points on the same line have the same total rates.

### A.2.3 RESULTS FOR GENERALIZED TOP-DOWN HVAEs ON CIFAR-10

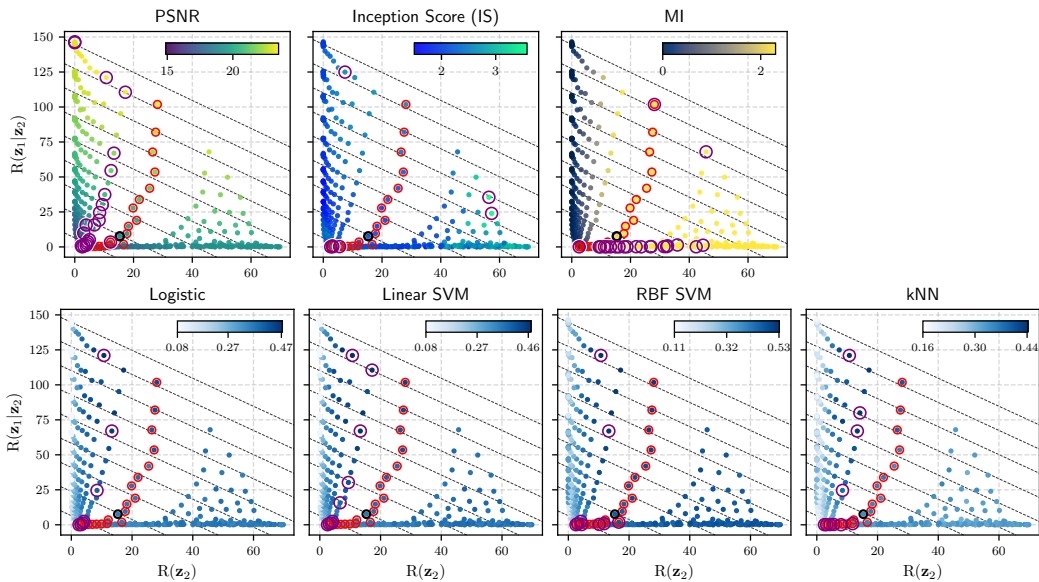

Figure 10: Trade-offs between rates and all metrics we used in Section 5 from the generalized top-down HVAEs trained with CIFAR-10. The results from the standard VAE (i.e. $\beta_2 = \beta_1 = 1$) and the $\beta$-VAE (i.e. $\beta_2 = \beta_1$) are marked with "o" and "o". The markers "o" highlight the optimal models selected using convex hull (see Figure 6 for details). The diagonal grid lines are references for equivalent total rates, i.e. points on the same line have the same total rates.

## B    PROOF OF THE BOUND ON CLASSIFICATION ACCURACY

This section provides a proof of Eq. 11 by reformulating the proof of Proposition 5 in the thesis by Meyen (2016) into the notation used in the present paper. We stress that this section contains no original contribution and is provided only as a convenience to the reader, motivated by reviewer feedback. All credits for this section belong to Meyen (2016).

We consider an (unknown) true data generative distribution $p_{\text{data}}(y, \boldsymbol{x})$ for data $\boldsymbol{x}$ with (unobserved) true labels $y$, and a hierarchical VAE with an inference model $q_\phi(\{\boldsymbol{z}_\ell\} \,|\, \boldsymbol{x})$ of the form of Eq. 4. Focusing on a single layer $\ell$ of latents, we denote the joint probability over $y$, $\boldsymbol{x}$, and $\boldsymbol{z}_\ell$ as

$$q(y, \boldsymbol{x}, \boldsymbol{z}_\ell) := p_{\text{data}}(y, \boldsymbol{x}) \, q_\phi(\boldsymbol{z}_\ell | \boldsymbol{x}) \tag{13}$$

where the marginal $q_\phi(\boldsymbol{z}_\ell | \boldsymbol{x})$ of $q_\phi(\{\boldsymbol{z}_\ell\} \,|\, \boldsymbol{x})$ is defined as usual. We further consider a classifier $p_{\text{cls.}}(y | \boldsymbol{z}_\ell)$ that operates on $\boldsymbol{z}_\ell$. Denoting its top prediction as $\hat{y} := \arg\max_y p_{\text{cls.}}(y | \boldsymbol{z}_\ell)$, the classification accuracy is $\alpha := \mathbb{E}_q[\delta_{y,\hat{y}}]$, where $\delta$ is the Kronecker delta.

**Theorem 1.** *The mutual information $I_q(y; \boldsymbol{z}_\ell)$ between the latent representation $\boldsymbol{z}_\ell$ and the true label $y$ under the distribution $q$ defined in Eq. 13 is lower bounded as follows,*

$$I_q(y; \boldsymbol{z}_\ell) \geq f(\alpha) \qquad \text{with} \qquad f(\alpha) = H_{p_{\text{data}}}[y] - H_2(\alpha) - (1 - \alpha) \log(M - 1) \tag{14}$$

*where $H_2(\alpha) = -\alpha \log \alpha - (1 - \alpha) \log(1 - \alpha)$ is the entropy of a Bernoulli distribution, $H_{p_{\text{data}}}[y] \leq \log M$ is the marginal entropy of the true labels, and $M$ denotes the number of classes.*

Before we prove Theorem 1, we note that the function $f$ is strictly monotonically increasing on the relevant interval $[\max_y p_{\text{data}}(y), 1]$. Thus, $f$ is invertible and we obtain the following corollary:

**Corollary 1.** *The classification accuracy $\alpha$ is upper bounded as in Eq. 11 of the main text, i.e.,*

$$\alpha \leq f^{-1}(I_q(y; \boldsymbol{z}_\ell)) \leq f^{-1}\big(\mathbb{E}_{p_{\text{data}}(\boldsymbol{x})}[R(\boldsymbol{z}_{\geq \ell})]\big). \tag{15}$$

The second inequality in Eq. 15 results from the bound $I_q(y; \boldsymbol{z}_\ell) \leq \mathbb{E}_{p_{\text{data}}(\boldsymbol{x})}[R(\boldsymbol{z}_{\geq \ell})]$ derived in Eq. 10, using the fact that $f^{-1}$ is monotonically increasing (since $f$ is).

*Proof of Theorem 1.* We split the mutual information into two contributions,

$$I_q(y; \boldsymbol{z}_\ell) = H_{p_{\text{data}}}[y] - H_q[y | \boldsymbol{z}_\ell] = H_{p_{\text{data}}}[y] - \mathbb{E}_{\boldsymbol{z}_\ell \sim q(\boldsymbol{z}_\ell)}\big[\mathbb{E}_{y \sim q(y | \boldsymbol{z}_\ell)}[-\log q(y | \boldsymbol{z}_\ell)]\big] \tag{16}$$

where, as clarified in the second equality, $H_q[y | \boldsymbol{z}_\ell]$ is the expectation over $\boldsymbol{z}_\ell$ of the conditional entropy of $y$ given $\boldsymbol{z}_\ell$, and $q(\boldsymbol{z}_\ell)$ and $q(y | \boldsymbol{z}_\ell)$ are marginals and conditionals of $q$ (Eq. 13) as usual.

Since $H_{p_{\text{data}}}[y]$ is fixed by the problem at hand, finding a lower bound on $I_q(y; \boldsymbol{z}_\ell)$ for a given classification accuracy $\alpha$ is equivalent to finding an upper bound on the second term on the right-hand side of Eq. 16, $H_q[y | \boldsymbol{z}_\ell] = \mathbb{E}_{\boldsymbol{z}_\ell \sim q(\boldsymbol{z}_\ell)}[\mathbb{E}_{y \sim q(y | \boldsymbol{z}_\ell)}[-\log q(y | \boldsymbol{z}_\ell)]]$, with the constraint $\mathbb{E}_q[\delta_{y,\hat{y}}] = \alpha$. We do this by upper bounding the conditional entropy $\mathbb{E}_{y \sim q(y | \boldsymbol{z}_\ell)}[-\log q(y | \boldsymbol{z}_\ell)]$ of $y$ given $\boldsymbol{z}_\ell$ for all $\boldsymbol{z}_\ell$ independently, and then taking the expectation over $\boldsymbol{z}_\ell \sim q(\boldsymbol{z}_\ell)$.

For a fixed latent representation $\boldsymbol{z}_\ell$, we first split off the contribution to $\mathbb{E}_{y \sim q(y | \boldsymbol{z}_\ell)}[-\log q(y | \boldsymbol{z}_\ell)]$ from $y = \hat{y}$, where $\hat{y} = \arg\max_y p_{\text{cls.}}(y | \boldsymbol{z}_\ell)$ is the label that our classifier would predict for $\boldsymbol{z}_\ell$,

$$\mathbb{E}_{y \sim q(y | \boldsymbol{z}_\ell)}[-\log q(y | \boldsymbol{z}_\ell)] = -q(y \!=\! \hat{y} | \boldsymbol{z}_\ell) \log q(y \!=\! \hat{y} | \boldsymbol{z}_\ell) - \sum_{y \neq \hat{y}} q(y | \boldsymbol{z}_\ell) \log q(y | \boldsymbol{z}_\ell). \tag{17}$$

Here, the second term on the right-hand side resembles the entropy of a distribution over the remaining $(M - 1)$ labels ($y \neq \hat{y}$), except that the probabilities sum to $(1 - q(y \!=\! \hat{y} | \boldsymbol{z}_\ell))$ rather than one. Thus, regardless of the value of $q(y \!=\! \hat{y} | \boldsymbol{z}_\ell)$, this term is maximized if $q(y | \boldsymbol{z}_\ell)$ distributes the remaining probability mass $(1 - q(y \!=\! \hat{y} | \boldsymbol{z}_\ell))$ uniformly over the remaining $(M - 1)$ labels, i.e.,

$$\mathbb{E}_{y \sim q(y | \boldsymbol{z}_\ell)}[-\log q(y | \boldsymbol{z}_\ell)] \leq -q(y \!=\! \hat{y} | \boldsymbol{z}_\ell) \log q(y \!=\! \hat{y} | \boldsymbol{z}_\ell) - (1 - q(y \!=\! \hat{y} | \boldsymbol{z}_\ell)) \log \frac{1 - q(y \!=\! \hat{y} | \boldsymbol{z}_\ell)}{M - 1}$$

$$= H_2(q(y \!=\! \hat{y} | \boldsymbol{z}_\ell)) + (1 - q(y \!=\! \hat{y} | \boldsymbol{z}_\ell)) \log(M - 1). \tag{18}$$

Plugging Eq. 18 back into Eq. 16, we obtain the bound

$$I_q(y; \boldsymbol{z}_\ell) \geq H_{p_{\text{data}}}[y] - \mathbb{E}_{\boldsymbol{z}_\ell \sim q(\boldsymbol{z}_\ell)}\big[H_2(q(y \!=\! \hat{y} | \boldsymbol{z}_\ell))\big] - \mathbb{E}_{\boldsymbol{z}_\ell \sim q(\boldsymbol{z}_\ell)}\big[1 - q(y \!=\! \hat{y} | \boldsymbol{z}_\ell)\big] \log(M - 1). \tag{19}$$

We arrive at the proposition (Eq. 14) by pulling the concave function $H_2$ out of the expectation using Jensen's inequality, and by then identifying $\mathbb{E}_{\boldsymbol{z}_\ell \sim q(\boldsymbol{z}_\ell)}[q(y \!=\! \hat{y} | \boldsymbol{z}_\ell)] = q(y \!=\! \hat{y}) = \alpha$.  □

