# OpenReview forum: "Trading Information between Latents in Hierarchical Variational Autoencoders"
_ICLR.cc/2023/Conference — ICLR 2023 poster_

### Official Review · Reviewer_nhzt · 2022-10-18

**Confidence:** 4
**Correctness:** 4
**Technical Novelty And Significance:** 2
**Empirical Novelty And Significance:** 3
**Recommendation:** 8

**Clarity, Quality, Novelty And Reproducibility:**

Clarity - The paper is clear. Everything, from the setup, diagrams, and presentation of empirical findings, focuses around a clear message. Equations and other mathematical expressions are clearly defined. Plots are shown with clear color-coding, shapes, and labels. The authors separate various empirical findings into distinct sections, figures, and tables.

Quality - The paper appears to be high-quality. The authors take a known setting (hierarchical VAEs) and perform a set of analyses on benchmark datasets using accepted metrics for tracking performance on downstream tasks (PSNR, Inception Score, and classification accuracy). Special cases (i.e., VAEs and beta-VAEs) are shown within empirical findings. The analysis appears to be fairly thorough.

Novelty - The novelty of the paper is one of its weaker aspects. The primary novel contribution of the paper is in demonstrating that having separate beta hyperparameters for each level in hierarchical VAEs is beneficial when using these models for various applications. The main findings of the paper are in sweeping over two hyperparameter dimension rather than one. Admittedly, there is some novelty in this approach, as Lagrange factors have not been explored extensively in the setting of hierarchical models.

Reproducibility - The paper is likely reproducible. The results show clear trends across multiple datasets while varying hyperparameters. That is, many models were trained in order to present the results in the paper, and the trends appear to be fairly smooth and consistent. The authors also provided accompanying code with their submission, which will assist with reproducing the results.

**Strength And Weaknesses:**

**Strengths**
- **Clear presentation.** The authors clearly lay out the setting (hierarchical VAEs), three application domains (classification, generation, and reconstruction), and the main contribution (evaluating different beta trade-offs). Supporting diagrams in Figure 2 show the setup considered by the authors. Mathematical equations provide clear and consistent definitions of the quantities under consideration without introducing superfluous notation. And the experiment section is laid out with separate sections and plots for each setting.
- **Great plots / empirical analysis.** Beyond the clear presentation, the empirical findings are presented extremely well. It’s apparent that a significant amount of time went into creating these plots. The plots present findings with a mix of color-coding, shapes, and 2D and 3D visualizations. The plots do a great job of presenting the distinction between VAE vs. beta-VAE vs. having multiple beta hyperparameters.
- **Somewhat practical/relevant.** The paper tackles a rather practical and straightforward question of whether adjusting the rate at each hierarchical level in VAEs significantly impacts performance on various applications. This is a well-defined question, which the authors directly address by sweeping over hyperparameters (betas) and evaluating performance using relevant metrics. While I’m not sure whether a reader would come away from the paper with clear guidance on how to adjust these hyperparameters for any given architecture, the main message—that these hyperparameters can and should be separately adjusted—is clear.

**Weaknesses**
- **Limited novelty.** The primary novel aspect of the paper is in proposing to separately adjust different beta hyperparameters for each level in hierarchical VAEs, showing that this is relevant for various applications. I did not find this to be particularly novel, as the paper boils down to performing a hyperparameter sweep over multiple (in this case, two) dimensions rather than one. Likewise, the authors do not propose any form of method for automatically adjusting these hyperparameters for a given application. Rather, the paper presents a set of hyperparameter sweeps, showing that different regions are relevant for different applications. It’s unclear how these findings can readily be extended to hierarchies of arbitrary depth beyond simply performing hyperparameter sweeps in these larger spaces. The primary way of fixing this weakness would be to demonstrate some new capability afforded by this technique and/or some new method for targeting a particular application. For instance, rather than separately sweeping over both beta hyperparameters, one might consider using the constrained optimization formulation presented by Rezende & Viola, 2018, in which a target rate (or distortion) is satisfied.
- **Simple-ish models and datasets, which may not be relevant for practical settings.** The authors explore their empirical question in the setting of somewhat small 2-level hierarchical VAEs using MNIST, SVHN, and CIFAR10, all of which are fairly small image datasets. While I entirely understand the need to manage the complexity of the empirical setting to make such analyses tractable, it does raise questions of whether these analyses offer practical guidance for a practitioner. I would have a hard time believing that any of the models and datasets in this paper would be a go-to choice for any of the applications (classification, generation, reconstruction) considered. I understand that the authors are not attempting to obtain SOTA performance in any of these domains. However, to address this weakness, it may be useful to target one particular application more directly, e.g., compression/reconstruction, and show that these findings are relevant with full-scale models and datasets.
- **Framing may not be fully general/applicable.** The authors consider a stripped-down version of hierarchical VAEs, in which the generative process is a strict Markov chain. However, such models are not typically considered within the literature. Starting with the IAF (Kingma et al., 2016) hierarchical VAE architecture, most hierarchical models have introduced deterministic connections between higher-level variables and lower-levels (including the conditional likelihood). This is true of modern large-scale hierarchical VAEs like NVAE (Vahdat & Kautz, 2020) and VDVAE (Child, 2021). These deterministic connections allow gradients to flow directly from the reconstruction to the higher-level variables, which undoubtedly alters the gradients and learning dynamics. I understand that the authors restricted their analysis to Markov hierarchies for practical purposes, but the findings may be moot if they are not relevant in the more general setting of non-Markov hierarchies. Fixing this weakness would involve performing analyses in this more general setting.

**Summary Of The Paper:**

The authors propose to assign separate beta Lagrange factors to each level in hierarchical VAEs to control the rate for each level separately. They argue that different applications of VAEs have different trade-offs for the rate, e.g., classification, generation, and reconstruction. The experiments sweep over values of each beta in 2-level VAEs on MNIST, SVHN, and CIFAR10. They report various metrics for each setting, showing that different values of beta are useful in different settings.

**Summary Of The Review:**

While the paper does not present any radically novel techniques or entirely surprising findings, the authors tackle a well-defined and practical question that, to the best of my knowledge, has not been addressed previously in the literature. The clarity of the presentation makes the paper approachable by readers with varying levels of background knowledge, and the empirical findings demonstrate a high degree of rigor (i.e., performing a large-scale sweep, multiple datasets, multiple metrics). For these reasons, I recommend acceptance.

---

> ### Author Response · Authors · 2022-11-18
> **Reply to Reviewer nhzt (R4) from the authors**
>
> Thank you for your comprehensive feedback! We are pleased that you found our work is “approachable by readers with varying levels of background”, it addresses “a well-defined and practical question” and that our “empirical findings demonstrate a high degree of rigor”.
>
> To incorporate your feedback, we address the concerns below and updated the PDF accordingly.
>
> > **Q1** - “It’s unclear how these findings can readily be extended to hierarchies of arbitrary depth beyond simply performing hyperparameter sweeps in these larger spaces.”
>
> We agree it is challenging to perform similar unbiased evaluations using grid search with hierarchies $L>2$. Luckily, many of the plots in Section 5 and the resulting guidelines naturally translate to $L>2$ as one may still choose to use only _two_ distinct $\beta$s in a well-motivated way.
>
> For example, consider a VAE with $L>2$ and a downstream classifier, which could now operate on any level $\mathbf z_\ell$ where $\ell\in\{1,\ldots,L\}$. Following the discussion in Section 5.6, one can then consider $R(\mathbf z_L)+\ldots+R(\mathbf z_\ell|\mathbf z_{\geq\ell+1})$ as the total information content accumulated up to representation $\mathbf z_\ell$ that is used for classification (corresponding to $R(\mathbf z_2)$ in Section 5.6); the total rate added after the classification layer is then $R(\mathbf z_{\ell-1}|\mathbf z_{\geq\ell})+\ldots+R(\mathbf z_1|\mathbf z_{\geq2})$ (this corresponds to $R(\mathbf z_1|\mathbf z_2)$ in the discussion of Section 5.6). Thus, our guidelines suggest that an expressive classifier requires high $R(\mathbf z_L)+\ldots+R(\mathbf z_\ell|\mathbf z_{\geq\ell+1})$ and low $R(\mathbf z_{\ell-1}|\mathbf z_{\geq\ell})+\ldots+R(\mathbf z_1|\mathbf z_{\geq2})$ whereas, for a less expressive (e.g., linear) classifier, the opposite region is favorable. In fact, a (crude) initial hyperparamter optimization could optimize over only _two_ distinct $\beta$-parameters by setting $\beta_L=\ldots=\beta_\ell$ and $\beta_{\ell-1}=\ldots=\beta_1$. This would still allow trading off between information content accumulated in $\mathbf z_\ell$ and information content added later. We added this as a comment to Section 3.
>
> > **Q2** - “rather than separately sweeping over both beta hyperparameters, one might consider using the constrained optimization formulation presented by Rezende & Viola, 2018 [...]”
>
> Thank you for recommending this interesting paper! The method proposed there could indeed be used to target specific layer-wise rates. We added a reference to this paper to our conclusions.
>
> As you correctly pointed out elsewhere, our paper shows that “different regions [in $\beta$-space] are relevant for different applications”. Our analysis identifies a _region_ in $\beta$-space where one can expect best performance for each task (see also added plots in lower right corner of Figure 1) but does not provide _precise_ values for optimal $\beta$s. In the end, the $\beta$s can be optimized with any hyperparameter optimization method in the literature, which is orthogonal to our contribution.
>
> > **Q3** - “Simple-ish models and datasets, which may not be relevant for practical settings. [...] it may be useful to target one particular application more directly [...]”
>
> We agree that the models and data sets in our empirical analysis are rather small due to computational limitations. We wanted to keep the empirical analysis as unbiased as possible, which we could only achieve with a grid search over hyperparameter space. Rather than investing a lot of computational resources into limited experiments on a larger model, our approach was to complement the empirical analysis in Section 5 with theoretical results (Section 4) for all three considered application domains, which apply to models of any size. We hope that the combination of thorough empirical results for two small models and three small data sets together with theoretical results for arbitrary model sizes provides sufficient confidence in the generality of our findings.
>
> > **Q4** - “Most hierarchical models have introduced deterministic connections between [high and low levels] like NVAE and VDVAE [but] the authors restricted their analysis to Markov hierarchies [...]”
>
> Thank you for pointing this out! Both NVAE and VDVAE use the same conditioning structure as first formulated in LVAE by Sønderby et al. In the appendix, we show results for LVAE, which are consistent with the findings in the main paper.
>
> We agree that the theoretical analysis in the initial version of the paper oversimplified the model architecture by making the additional Markov chain assumption. However, this assumption was only done to simplify the notation and is in fact not necessary. We carefully checked and updated all definitions and derivations in our paper and uploaded a new revision of the PDF that removes the unnecessary Markov chain assumption.
>
> We would appreciate it if you could let us know whether our above responses resolve your concerns.

---

> > ### Comment · Reviewer_nhzt · 2022-12-05
> > **Reply to authors**
> >
> > Thank you for the response. After reading the other reviewers comments and the authors' responses, I am still satisfied with the scope and content of the paper. Unless the other reviewers have specific concerns that they feel detract from the paper, I intend to maintain my score.

---

### Official Review · Reviewer_7gHR · 2022-10-24

**Confidence:** 3
**Correctness:** 2
**Technical Novelty And Significance:** 3
**Empirical Novelty And Significance:** 2
**Recommendation:** 5

**Clarity, Quality, Novelty And Reproducibility:**

The paper is well written, although the assumptions of some statements are unclear, such as Eq 11. I think the idea of using different $\beta$ for HVAE is straightforward, not novel. On the other hand, the numerical experiments that show the different tasks require different $\beta$ in Figure 1 are important in the field.

**Strength And Weaknesses:**

# Strength
- This paper explicitly discusses how the three commonly used tasks of VAEs are related to the rates in the IB formulation. Then the authors showed theoretically and empirically that the different rates are preferable for different tasks, which seems missed in existing work.
- The authors extend the IB formulation of VAE to the different levels of regularization of rates in HVAE.
- The authors studied how the rates affect the performance of different tasks by controlling the regularization in detail.

# Weakness
- I am not satisfied with the explanation of how the rates of the representation $Z$ and accuracies in downstream tasks are related to each other. In particular, the authors introduced the bound Eq.11, but I do not think the assumptions of Eq. 11 in Meyen 2016 are satisfied in VAEs.
- Decomposing the rate term in IB and introducing different $\beta$s in HVAEs are natural. Although using different $\beta$s introduces additional flexibility and improves the performance, I do not think it is practical because we must choose $\beta$s, and it is not clear how to tune them.
- As the authors mentioned in Sec 6, it would be better to include the results about how we choose $\beta$s or the results that show the merit of using $\beta_1\neq \beta_2$ not $\beta_1=\beta_2$. In the current numerical experiments, I could not see that advantage.

**Summary Of The Paper:**

This paper clarifies that the different applications in VAEs require different trade-offs in the IB formulation; that is, the single regularization coefficient of rates in IB cannot achieve the best performance in three application domains. Motivated by this analysis, the authors proposed to extend the existing HVAEs based on the IB formulation and introduce flexible regularization terms in HVAEs. The new model can show better trade-offs under different tasks thanks to this flexibility.

**Summary Of The Review:**

I think the contributions of this work are two folds;
one is the analysis that the different tasks in VAEs require different levels of regularization about rates in IB problems. The other contribution is introducing the new HVAE that has flexible rate regularization.

I think the first contribution is insightful and should be shared in the community. Still, the second contribution seems insufficient since I could not be convinced why using different beta is important numerically or theoretically.
It would be better to include a discussion about how using different betas leads to resolving the problem of the necessity of different rates among different tasks by introducing additional flexibility.

---

> ### Author Response · Authors · 2022-11-10
> **Reply to Reviewer 7gHR (R3) from the authors**
>
> Thank you for your detailed feedback! We are encouraged that you found our analysis of different tasks for VAEs “insightful”, that you think it “should be shared in the community”, and that the numerical results “are important in the field”. We address your concerns below and updated the PDF accordingly.
>
> > **Q1** - “I do not think the assumptions of Eq. 11 in Meyen 2016 are satisfied”
>
> We understand that this could be confusing since our paper uses very different notation than Mayen (2016). We therefore uploaded a new revision with an added Appendix B, which rephrases the proof in our notation.
>
> Here's a brief informal explanation: the mutual information $I_q(y;\mathbf z_\ell)$ measures how much information about the true label $y$ is contained in the latent $\mathbf z_\ell$. For example, if $I_q(y;\mathbf z_\ell)$ is just barely above zero, then then $y$ and $\mathbf z_\ell$ are almost statistically independent and a classifier on $\mathbf z_\ell$ cannot do much better than random guessing. Aside from the formal proof added in Appendix B, we obtain additional confidence in the claim from Figure 6, which verifies the bound in an empirical example, and also shows that it is indeed a nontrivial bound.
>
> If the above informal explanation or the formal proof in Appendix B does not clarify the issue, then we would like to ask for more details about your concerns, and we are happy to address them. If the issue is resolved then we would appreciate if you could reconsider your correctness score of our paper.
>
> > **Q2** - “Decomposing the rate term in IB and introducing different $\beta$s in HVAEs are natural. [...] straightforward, not novel.”
>
> We would like to refer the reviewer to part (i) of our global comment titled “Clarification regarding novelty and guidelines for choosing $\beta$s”. In short, we agree that our method is simple, but we see this as an advantage, and we would like to stress that simplicity does not imply lack of novelty. Further, our paper goes beyond merely stating that the rate can be decomposed into layer-wise contributions: it provides a through theoretical and large-scale empirical analysis of the consequences of doing so, with practical advice on how to tune rates (or, equivalently, $\beta$s) for various tasks (see also next paragraph).
>
> > **Q3** - “we must choose $\beta$s, and it is not clear how to tune them”; “include the results about how we choose $\beta$s”
>
> While our work does not identify _precise_ values for optimal choices of the $\beta$s, it identifies _regions_ in $\beta$-space of good performance on various tasks, both theoretically (Section 4) and empirically (Section 5). We realize that the paper characterizes these regions mostly in rate-space rather than in $\beta$-space, but these two spaces are directly related (they are duals) and the guidelines on optimal rates provided in the paper can directly be translated into guidelines for setting the $\beta$s (see part (ii) of our global comment titled “Clarification regarding novelty and guidelines for choosing $\beta$s”).
>
> We also added new plots in the bottom right part of Figure 1, which show the performance landscapes of various tasks directly in $\beta$-space. For each task, one obtains near-optimal performance in an extended region in $\beta$-space. Thus, even approximate guidelines such as $\beta_2 >\beta_{1}$ can be useful in practice.
>
> > **Q4** - “include ... results that show the merit of using $\beta_{1}\neq\beta_2$ not $\beta_{1}=\beta_2$.”
>
> Our plots in Section 5 all indicate models with $\beta_{1}=\beta_2$ by red circles. The fact that these don't typically coincide with the optimal models for a specific task (purple circles in Figures 3, 5, and 7) shows the merit of setting $\beta_{1}\neq\beta_2$. The new plots in the lower right part of Figure 1 in the updated PDF also show the benefits of setting $\beta_{1}\neq\beta_2$.
>
> > **Summary** - “I could not be convinced why using different beta is important numerically or theoretically.”
>
> We hope that our reproduced proof of Eq. 11 in Appendix B resolves the concerns about theory (see answer to **Q1** above), and that our answers to **Q3** and **Q4** above resolve the concerns about numerical evidence.
>
> > **Summary (cont'd)** - “It would be better to include a discussion about how using different betas leads to resolving the problem of the necessity of different rates among different tasks by introducing additional flexibility.”
>
> If we understand the reviewer correctly, the concern is about how to find optimal values for the $\beta$s. We hope that this is addressed by our answer to **Q3** above.
>
> With the the new plots and the added proof, we think that the updated version of the paper now provides readers with clearer guidance on how to tune the $\beta$s of hierarchical VAEs for given applications.
>
> We would appreciate it if you could let us know whether our above responses resolve your concerns, and change your assessment of our work accordingly.

---

### Official Review · Reviewer_kj2j · 2022-10-24

**Confidence:** 4
**Correctness:** 3
**Technical Novelty And Significance:** 2
**Empirical Novelty And Significance:** 2
**Recommendation:** 6

**Clarity, Quality, Novelty And Reproducibility:**

**Clarity & Quality**
* The plots are beautiful, but I had some trouble interpreting them. In particular, identifying the (many) overlapping colored circles was pretty challenging, and required me to zoom in heavily.  Also it may help to present a separate plot for each of the three main tasks considered.
* Writing quality is high.

**Novelty**
* The generalization of $\beta$-VAEs to multiple layers is very sensible and well-motivated, and the perspective to study the impact of $\beta$ values on different tasks was certainly interesting. However, I did not find much novelty in the paper -- the theoretical exposition consists mostly of relatively straightforward generalization of existing bounds, with the remainder of the paper being a thorough empirical study.

**Strength And Weaknesses:**

**Strengths**
* Very well-written paper with clear motivation, with an interesting generalization of $\beta$ VAEs.
* Thorough analysis and discussion.

**Weaknesses**
* Lack of empirical results for $L > 2$.  While I understand that it may not be feasible to perform a thorough grid search (as was done for the two-layer VAEs in this paper) with deeper VAEs, basing all empirical conclusions on $L=2$ experiments seems a little brittle.  While the theoretical discussions are sensible, it is unclear to me whether the empirical results presented would readily generalize to much larger values of $L$ -- especially given finite capacity models with possibly different optimization dynamics  in practice.



**Summary Of The Paper:**

This paper presents a relatively thorough study of the effect of $\beta$ for the case of hierarchical VAEs. Specifically, the authors study the rate-distortion performance of two-layer VAEs with various combinations of $\beta$s and show that different tasks require different settings of $\beta$s.  The paper also presents a theoretical discussion that predicts the empirical results.

**Summary Of The Review:**

Overall, I enjoyed reading the paper as it is very well-written and the theoretical discussions conclusions are coherent with the empirical findings.  I particularly value the fact that the paper explicitly tries to provide useful guidelines for practitioners who may need to tune a hierarchical VAE for a specific task.  However as mentioned above, the paper does not present a novel idea, and is somewhat incomplete in experiments.

---

> ### Author Response · Authors · 2022-11-18
> **Reply to Reviewer kj2j (R2) from the authors**
>
> Thank you for your insightful feedback! We are pleased that you “enjoyed reading the paper” and found our work “a thorough empirical study”! Most importantly, we are encouraged that you “value the fact that the paper explicitly tries to provide useful guidelines for practitioners”. This was indeed a central goal for us.
>
> To incorporate your feedback, we address the suggestions below and updated the PDF accordingly.
>
> > **Q1** - “Lack of empirical results for $L>2$. While I understand that it may not be feasible to perform a thorough grid search [...]”
>
> We agree that an analysis of VEAs with more than two layers of latents would be interesting but that it would be infeasible to do this in a similarly unbiased evaluation using grid search. It would also make plotting the results even more challenging. Luckily, many of the plots in Section 5 and the resulting guidelines naturally translate to $L>2$ as one may still choose to use only two distinct $\beta$-hyperparameters in a well-motivated way as follows.
>
> For example, consider a VAE with $L>2$ and a downstream classifier, which could now operate on any level $\mathbf z_\ell$ where $\ell\in\{1,\ldots,L\}$. For the purpose of the discussion in Section 5.6, one can then consider $R(\mathbf z_L)+\ldots+R(\mathbf z_\ell|\mathbf z_{\geq\ell+1})$ as the total information content accumulated in the representation $\mathbf z_\ell$ that is used for classification (corresponding to $R(\mathbf z_2)$ in the discussion of Section 5.6); the total rate added after the classification layer is then $R(\mathbf z_{\ell-1}|\mathbf z_{\geq\ell})+\ldots+R(\mathbf z_1|\mathbf z_{\geq2})$ (this corresponds to $R(\mathbf z_1|\mathbf z_2)$ in the discussion of Section 5.6). Thus, our guidelines suggest that an expressive classifier requires high $R(\mathbf z_L)+\ldots+R(\mathbf z_\ell|\mathbf z_{\geq\ell+1})$ and low $R(\mathbf z_{\ell-1}|\mathbf z_{\geq\ell})+\ldots+R(\mathbf z_1|\mathbf z_{\geq2})$ whereas, for a less expressive (e.g., linear) classifier, the opposite region is favorable. In fact, a (crude) initial hyperparamter optimization could optimize over only _two_ distinct $\beta$-parameters by setting $\beta_L=\ldots=\beta_\ell$ and $\beta_{\ell-1}=\ldots=\beta_1$. This would still allow trading off between information content accumulated in $\mathbf z_\ell$ and information content added later. We added this as a comment to Section 3.
>
> > **Q2** - “The plots are beautiful, but I had some trouble interpreting them. [...] overlapping colored circles”
>
> Good point! We realized in particular that the original color scheme in Figure 5 was unfortunate, and we changed it out for a better color scheme that makes the red and purple circles stand out more.
>
> In general, we are aware that the plots in Section 5 are quite dense. It is important to us that we show _all_ results from the large-scale grid searches so that there can be no cherry picking. At the same time, we realized that readers may need some clues to guide their focus when looking at the plots. We hope that the purple circles, which identify optimal models for a given total rate, help guide focus, and that they are now easier to see also in Figure 5 with the new color scheme.
>
> > **Q3** - “it may help to present a separate plot for each of the three main tasks considered.”
>
> Thank you for the suggestion. We assume that this refers to Figure 1. Separate plots for the three tasks are provided later in the paper, in Figures 3, 5, and 7. We realize that this connection may not be easy to see since Figure 1 and Figures 3, 5, and 7 are so far apart in the paper. Therefore, we followed your suggestion and added three additional plots directly into Figure 1, which also analyze each task separately. These additional plots have the added benefit of providing an empirical analysis directly in $\beta$-space rather than in the (dual) rate space of Figures 3, 5, and 7.
>
> > **Q4** - “not [...] much novelty [...] the remainder of the paper being a thorough empirical study.”
>
> We’d like to refer the reviewer to the separate thread titled “Clarification regarding novelty and guidelines for choosing $\beta$s”. In short, we agree that our method is simple and may seem straightforward in hindsight. But we see this as an advantage, and we would like to stress that simplicity does not imply lack of novelty. To the best of our knowledge, a similar analysis has not been reported before in the literature.
>
> We agree with the reviewer that a central contribution of our paper is the thorough empirical study. As stated by the reviewer in the summary, we believe that our evaluation provides “useful guidelines for practitioners”, and we think that these are of value to share with the community. In this context, the main purpuse of the theory part (Section 4) is to provide additional confidence that the empirical results are not just a coincidence.
>
> We would appreciate it if you could let us know whether our above responses resolve your concerns.

---

### Official Review · Reviewer_xknb · 2022-10-26

**Confidence:** 3
**Correctness:** 3
**Technical Novelty And Significance:** 2
**Empirical Novelty And Significance:** 2
**Recommendation:** 6

**Clarity, Quality, Novelty And Reproducibility:**

The writing of the paper is clear and easy to follow.


**Strength And Weaknesses:**

Strength:

The paper was well written with good structures.

Weakness

The paper is a simple extension of existing works, and the significance and novelty of the paper are not strong enough for publication at the moment. The paper could be stronger with additional formal theoretical study.



**Summary Of The Paper:**

The authors extend an existing method and develop an approach to control each layer’s contribution to the rate independently in the hierarchy VAE. They identify the most general class of inference models to which their proposed method is applicable. In the experiments, they demonstrate that the proposed method better tunes hierarchical VAEs.

**Summary Of The Review:**

The author could add more theoretical and experimental studies to make the paper stronger. For example, how does the distribution of $\beta$s affect the performance? How many layers of H-VAEs do we need given specific datasets?

---

> ### Author Response · Authors · 2022-11-18
> **Reply to Reviewer xknb (R1) from the authors**
>
> Thank you for your valuable feedback! We address each concern below.
>
> > **Q1** - “The paper is a simple extension of existing works, and the significance and novelty of the paper are not strong enough for publication at the moment.”
>
> We would like to refer the reviewer to part (i) of our global comment titled “Clarification regarding novelty and guidelines for choosing $\beta$s”. In short, we agree that our method is simple, but we see this as an advantage, and we would like to stress that simplicity does not imply lack of novelty. Further, we consider it an important contribution to not only state that the rate can be decomposed into layer-wise contributions, but to also provide a through theoretical and large-scale empirical analysis of the consequences of doing so; as well as practical advice on how to tune rates (or, equivalently, $\beta$s) for various tasks (see also next paragraph). Given the popularity of VAEs, we argue that such advice can be very significant for practitioners.
>
> > **Q2** - “The author could add more theoretical and experimental studies to make the paper stronger. For example, how does the distribution of $\beta$s affect the performance?”
>
> The discussion of how the choice of $\beta$s affects performance makes up the main part of our paper. We address this issue theoretically in Section 4 and empirically in Section 5. For example, Figures 3, 5, and 7 show performance metrics for the three application domains of VAE as a function of layer-wise rates.
>
> We realize that the performance analysis in the paper is expressed mostly in rate-space rather than in $\beta$-space, but these two spaces are directly related (they are duals) and the practical guidelines on optimal rates provided in the paper can directly be translated into guidelines for setting the $\beta$s (see part (ii) of our global comment titled “Clarification regarding novelty and guidelines for choosing $\beta$s”).
>
> We also added new plots in the bottom right part of Figure 1, which show the performance landscapes of various tasks directly in $\beta$-space. For each task, one obtains near-optimal performance in an extended region in $\beta$-space. Thus, even approximate guidelines such as $\beta_2>\beta_1$ can be useful in practice.
>
> > **Q3** - “How many layers of H-VAEs do we need given specific datasets?”
>
> The optimal number of latent layers $L$ for HVAEs given specific datasets can be tuned like other hyperparameters, e.g., with Bayesian optimization, which is an interesting problem for HVAE itself and not specific to our proposed method.
>
> We emphasize that the focus of our paper is not to propose a new model architecture for hierarchical VAEs. Such proposals exist plentiful in the literature. Our contribution is orthogonal to the model architecture: rather than proposing a specific architecture, we propose and analyze a new method to train and tune HVAEs that applies to a large class of popular HVAE architectures, and to an arbitrary number of layers.
>
> We would appreciate it if you could let us know whether our above responses resolve your concerns, and change your assessment of our work accordingly.

---

> > ### Comment · Reviewer_xknb · 2022-11-23
> > **Response**
> >
> > After reading the authors' responses to the reviews, I would like to update the score. Although the paper's novelty is still a weakness, the study is solid with experiments.

---

### Author Response · Authors · 2022-11-09
**Clarification regarding novelty and guidelines for choosing $\beta$s**

Dear reviewers,

Thank you very much for your insightful comments! Before addressing each review individually over the next few days, please allow us to quickly address two overall issues:

### (i) Novelty
We fully agree with the reviewers that our proposed information trading framework in hierarchical VAEs is very simple and might even appear obvious in hindsight. However, simplicity does not imply lack of novelty and, to the best of our knowledge, the presented idea and a thorough empirical analysis of it has not been reported before. This claim is also supported by comments from reviewers, e.g., _“... which seems missed in existing work_ **_(R3)_**_”_, _“... has not been addressed previously in the literature_ **_(R4)_**_”_.

We strongly believe that the simplicity of our proposed method is an asset rather than a shortcoming, because it makes the method widely applicable and accessible to practitioners, whose main expertise often lies in fields other than information theory or Bayesian inference.

**Updated PDF:** We understand that the original abstract might have set the wrong expectations, and we thus slightly rephrased it to be upfront about the simplicity of the proposed method.

### (ii) Connection Between Rates and $\beta$s
We share the view with reviewers that our current work does not provide an algorithm for identifying the _precise_ optimal $\beta$s for a given task. However, we do identify _regions_ in $\beta$-space where one should expect good performance, thus providing _“useful guidelines for practitioners_ **_(R2)_**_”_.

After reading the reviews, we realized that our original guidelines may not seem actionable at first sight as they are often phrased in terms of rates rather than $\beta$s (because rates admit a direct information theoretical interpretation). However, the rate-guidelines directly translate to $\beta$-guidelines as follows. The plots in Section 5 all identify the models with $\beta_{1}=\beta_{2}$ as a line of red circles, and most of our guidelines state that optimal models (purple circles in Figures 3, 5, and 7) lie either above or below this line, which is equivalent to $\beta_{1} < \beta_{2}$ or $\beta_{1} > \beta_{2}$, respectively.


**Updated PDF:** To communicate the implied $\beta$-guidelines more clearly, we will insert a clarification of the form “i.e., $\beta_{1} > \beta_{2}$” or “i.e., $\beta_{1} < \beta_{2}$” wherever we make statements that compare rates. We also just uploaded a new revision that adds three plots to the bottom right of Figure 1 which show three performance metrics directly as a function of $\beta_{1}$ and $\beta_2$.

We look forward to hearing back from you whether this comment clarifies the above two issues. We will address the remaining issues raised by individual reviewers in separate comments.

---

### Author Response · Authors · 2022-11-18
**Summary of updates in the PDF revision - [End of Discussion Stage 1]**

We would like to thank the reviewers for their time and valuable suggestions on our submission! We believe these suggestions significantly improve the paper. Here, we summarize the major changes in our PDF that resulted from the discussion:

- **Abstract:** slightly rephrased to be upfront about the simplicity of the proposed method.
- **Section 1:** added three plots to the bottom right of Figure 1 which show the three performance metrics directly as a function of $\beta_1$ and $\beta_2$.
- **Section 3.2:** added comment addressing that one may still choose to use only two distinct $\beta$s for $L>2$.
- **Sections 3 and 4:** removed the (unnacessary) constraint to a Markovian generative model; this forced us to slightly change the notation.
- **Section 5 and Appendix A.2:** better color code for Inception Score (e.g., Figure 5).
- **Section 6:** made the connection between targeting specific layer-wise rates and method from Rezende & Viola (2018)
- **Appendix B:** explicit proof of Eq. 11 (reformulates the proof in Meyen (2016) in our notation).

We hope that our responses in stage 1 have resolved the reviewers' concerns. We are looking forward to further discussions.

---

### Decision · Program_Chairs · 2023-01-20

**Decision:**

Accept: poster

**Justification For Why Not Higher Score:**

Lacking novelty in the theoretical part and hierarchies with more stochastic layers should be considered.

**Justification For Why Not Lower Score:**

One could argue for reject, but especially the empirical investigation is of a high quality that will be useful to see at ICLR.

**Metareview: Summary, Strengths And Weaknesses:**

This paper is about a bit-distortion rate analysis of hierarchical (beta)VAEs and using control of the bit-distortion rate for controlling the behavior and interpret each layer of the model.

The reviewers like the paper although they appreciate the experiments more than the novelty. Some also complain about that the hierarchy is maximum two stochastic layers.

**Note From Pc:**

if the above contains the word "oral" or "spotlight" please see: "oral" presentation means -> notable-top-5% and "spotlight" means -> notable-top-25%. As stated in our emails, we are disassociating presentation type from AC recommendations